# Net widening of Southern California beaches

Jonathan A. Warrick [1] ✉, Kilian Vos[2], Daniel D. Buscombe[1,3], Andrew C. Ritchie[1], Sean Vitousek [1], Teresa Hachey [4] & Brett F. Sanders [4]

Human impacts from dams reduce river sediment fluxes and are primary causes of coastal erosion worldwide. Here we provide new satellite-derived shoreline observation techniques to examine beach area trends across the diverse coastal settings of California. Contrary to global trends, these data reveal that the most heavily urbanized and dammed region of southern California experienced net beach growth of over 2 million m$^2$ during 1984-2024. While several beaches experienced severe erosion, overall widening is explained by sufficient sediment supply and concentrated widening from longshore transport captured at coastal structures and in littoral convergence zones. These results indicate that adequate sediment sources exist in this human-modified landscape to mitigate coastal erosion, but that this sediment is not effectively distributed to vulnerable beaches. This highlights the critical role that longshore sediment transport plays in long-term beach trends and illuminates management opportunities for coastal sustainability at the regional scale.

Beaches of the world's coasts provide important economic benefits[1], recreational opportunities[2], and habitats for ecosystems[3]. Beaches are also dynamic landscapes that change over time in response to waves[4], sea levels[5], sediment supplies[6,7], geologic setting[8], and human-built modifications to the coast[9]. The future of coastal landforms, including beaches, will depend on a range of local-to-global scale factors, including sea-level rise and sediment supplies, and the physical processes that move sediment to, along and away from these coastal landforms[9–12].

Human impacts from the construction of dams have reduced river sediment discharge to the coast throughout the world, increasing the potential for coastal degradation and shoreline erosion[11,13–16]. Land use changes, including the urbanization of coastal watersheds, alter the timing, volume and characteristics, such as grain-size distributions, of sediment discharged to the coast[11,17–20]. During the modern era of population increase, land use change, water resource development, sea level rise and changing coastal storms, it is important to develop a better understanding of these human impacts on the sustainability of coastal resources, including beaches[21–23].

The highly modified landscape of California (Fig. 1a) provides an ideal study area to evaluate beaches and their response to human impacts, owing to the understanding that the region's coastal watersheds provide a dominant source of California's littoral sediment, and to its diversity of coastal settings, including a range of population densities, land use changes, and coastal modifications[24–27]. The southern portion of California (Fig. 1) includes the majority of the region's population, wide-spread urbanization, and large dams on all of its rivers, the latter of which are suggested to have reduced sand flux to the coast to approximately 50% of historical levels[24,28]. Scientific study of California's beaches also benefits from almost a century of coastal research in the region that has resulted in fundamental concepts and understanding of coastal morphodynamics and sediment movement through littoral cells[9,25,29–32]. California's beaches are also valuable natural resources, and the environmental and economic costs associated with the erosion of these beaches would be profound[33–35].

Recently, it has been reported that most California beaches have transitioned to a state of chronic erosion during the past three-to-five decades and that this erosion will only increase in the future[27,36,37]. This transition appears to be consistent with the presumed effects of urbanization and dams on reducing coastal sediment supplies from the region's watersheds[24,37], even though field studies suggest that California's urban sediment supplies are much more complex than

[1]U.S. Geological Survey, Santa Cruz, CA, USA. [2]OHB Digital Services, Konrad-Zuse-Str. 8, Bremen, Germany. [3]Washington State Department of Ecology, Applied Coastal Research and Engineering, Olympia, WA, USA. [4]Department of Civil and Environmental Engineering, University of California, Irvine, CA, USA. ✉e-mail: jwarrick@usgs.gov

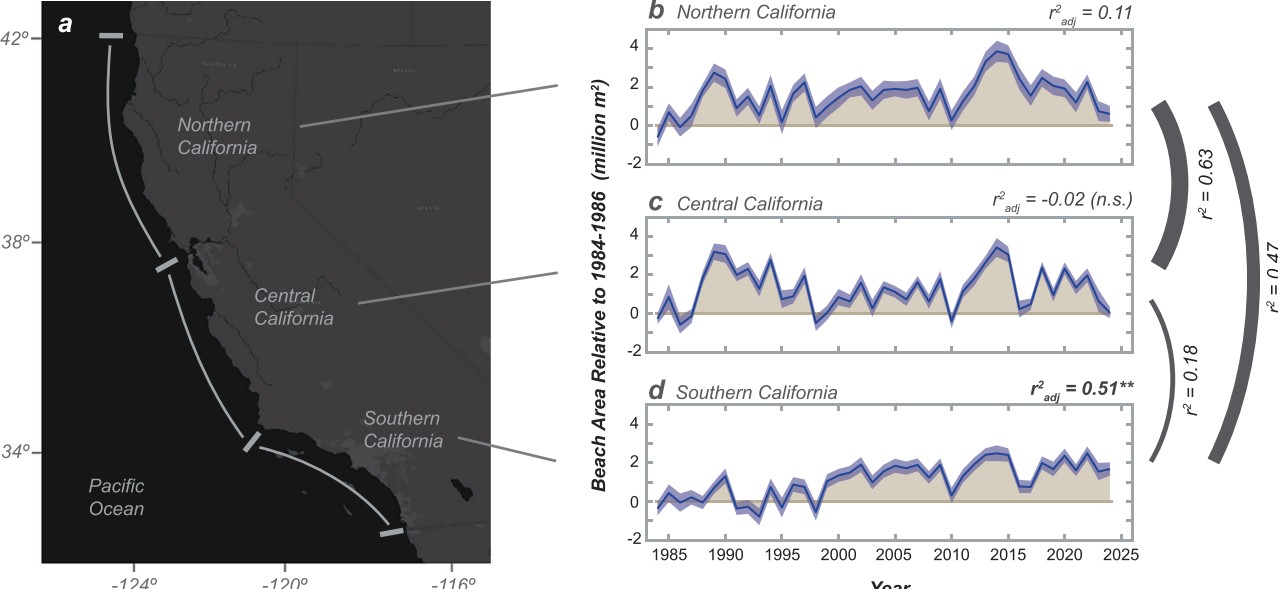

**Fig. 1 | Integrated beach change for the northern, central, and southern regions of California. a** Site map showing the three primary geographic regions of California for which integrated beach area changes were calculated. **b–d** Integrated beach area changes for northern, central, and southern California from the mean 1984–1986 value. Blue shading about the data represents 2-σ total uncertainty (refer to Methods). Tan shading represents the differences between annual values and the initial 1984–1986 mean value. For each region the adjusted correlation coefficient ($r^2_{adj}$) is provided for linear regression through the time series, which is denoted as highly significant ($p < 0.001$) with bold text and stars, significant ($0.001 < p < 0.05$) without notation, or not significant ($p > 0.05$) with text (n.s.) Additionally, the correlation coefficients between each regional time series and lines scaled to these coefficients are provided on the righthand side. Supplemental Fig. S1 shows these values converted into mean annual shoreline position changes (in meters). Base map from Ersi and its licensors.

these presumed effects[17,38]. This presumed erosional episode in southern California followed pervasive beach widening during the mid-20th century when excess sediment from harbor and other construction projects was disposed at the coast[27,32]. However, these recent findings of a shift to chronic erosion were developed from data with only 2-to-3 historical records of shoreline positions, each of which having large uncertainties[27,37]. Because these measurements directly influence estimations of the magnitude and costs of sediment nourishment, coastal engineering, and natural and human constructed resource losses related to coastal changes[34,37], these findings can influence hundreds of millions of US dollars in sediment management costs[39], billions of dollars of property and infrastructure, and millions of California residents[40]. It is critical, therefore, to ensure that coastal assessments are based on the best available information and integrated with supporting factors that influence shoreline change.

Here, we examine beach changes along the California coast to explore the effects of these human impacts on the status and trends of the region's numerous sandy beaches. We reanalyze the rates, patterns and behaviors of shoreline change across California with improved satellite-based measurements[41,42] and newly developed time-series analyses[43,44] to provide a reassessment of the trends and patterns in beach area change (refer to Methods). In doing so, we revise the assessments of human impacts on coastal change in the California study area and provide a new data synthesis technique that can be applied broadly to better understand coastal systems throughout the world. Most importantly, we find that the beaches of the southern California region have exhibited net widening—not chronic erosion—over the past four decades, which is the product of persistent widening of several beach segments in areas of net sediment convergence. This sediment convergence occurs in response to an abundance of sediment supply from fluvial sources, longshore transport, direct human management, and the coastal structures, such as harbors, breakwaters and jetties, that interrupt longshore transport near these beaches. These results will likely introduce new opportunities to redefine sand

management and redistribution strategies at local, littoral and regional scales to enhance long-term coastal resilience and sustainability.

## Results

### Regional changes along California

Over the regional scales of northern, central and southern California, we document strong year-to-year variations in shoreline positions, including erosion during 1995, 1998, 2010 and 2016 and recovery during subsequent years (Fig. 1), patterns that are consistent with field observations[45–48]. Additionally, there is reasonably strong correlation between the time series of beach areas of northern and central California ($r^2 = 0.63$), whereas correlations between southern California and the other regions are more moderate ($r^2 = 0.18$-47; Fig. 1) but still statistically significant. This provides evidence that there are temporally coherent phases of erosion and recovery across California but that the southern California shorelines have been exhibiting somewhat different patterns from those in central and northern California.

One of the key differences between the three regions of California is the rate of change, or trend, in shoreline positions and beach area over time. Notably, there is a highly significant positive trend (linear regression, $r^2_{adj} = 0.51$, $p < 0.001$) in beach area for southern California, whereas the trend in northern California is moderate to weak ($r^2_{adj} = 0.11$, $p = 0.02$) and there is no significant trend in central California ($r^2_{adj} = -0.02$; $p = 0.56$; Fig. 1). The net widening of northern California beaches, although barely decipherable owing to high year-to-year variability, is not unexpected, owing to the introduction of a massive accretionary wave of sediment in the late 1980s and early 1990s resulting in 1.2 million m² of new subaerial beach in the Klamath littoral cell[49]. This increase in beach area from newly introduced sediment scales with the weak trend in net beach growth across the entire northern California region (+1.2 ± 1.0 million m² of beach area over the 41-yr record; ±95% c.i.), which provides a link—albeit a statistically weak one—that the integrated changes in beach area may be associated with changes in littoral sediment volumes.

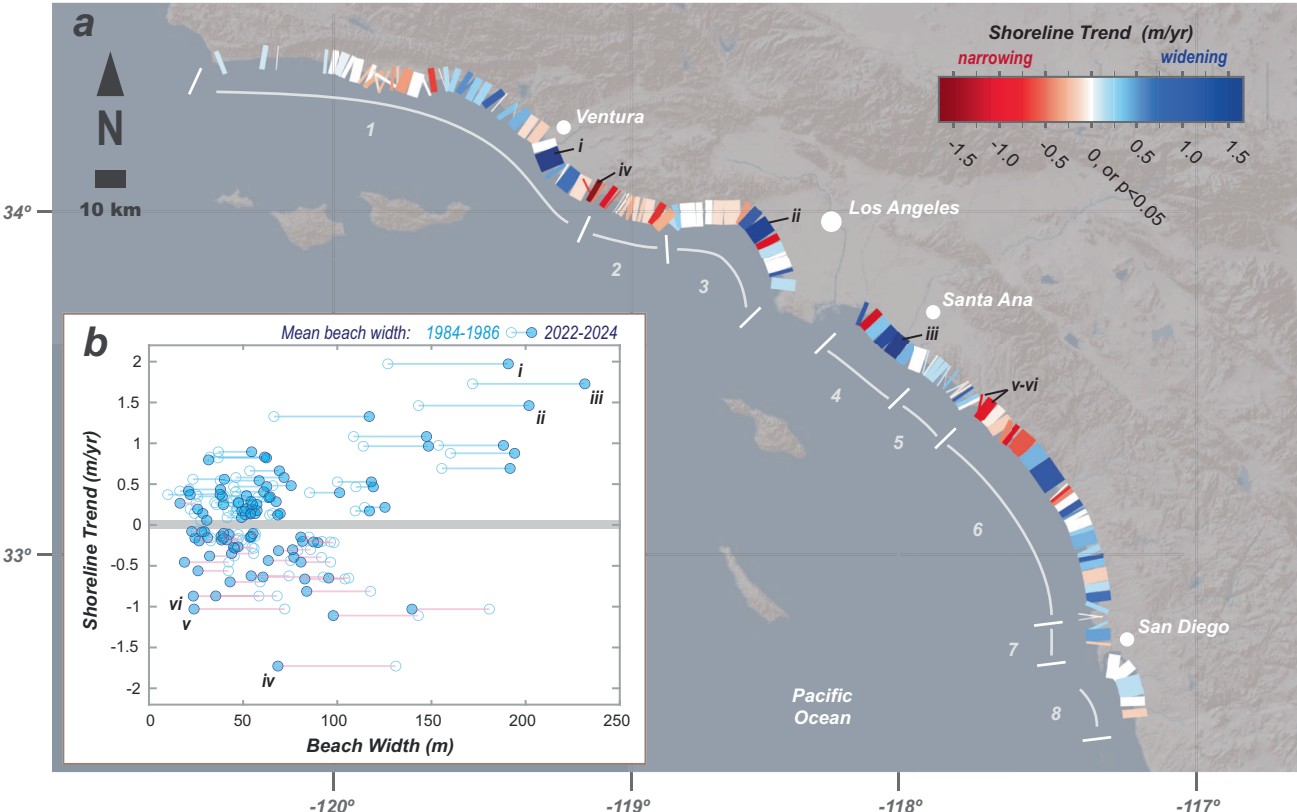

**Fig. 2 | Southern California shoreline trends at beach-segment scales.**
**a** Southern California littoral cells and rates of beach change for beach segments of this study during the 1984–2024 observations. Trends were computed by linear regression, and non-significant regressions ($p < 0.05$) are shown as zero trend. Littoral cells are shown with numbers, including: 1-Santa Barbara, 2-Zuma, 3-Santa Monica, 4-San Pedro, 5-Laguna, 6-Oceanside, 7-Mission Bay, 8-Silver Strand. Roman numerals (i-iii) denote the most rapidly widening beach segments (McGrath State Beach, Huntington Beach and Venice Beach) highlighted in Figs. 2b and 4, and roman numerals (iv-vi) denote rapidly narrowing beach segments (Mugu Lagoon Beach, Doheny Beach and Capistrano Beach) highlighted in Figs. 2b and 5. **b** Overall measured change in beach width during the observations compared to the shoreline trends for the beach segments. Beach width changes are defined as the difference between the first three years and the last three years of the study (1984–1986 and 2022–2024, respectively). Beaches with non-significant shoreline trends ($p < 0.05$) are not shown in (**b**). Base map from Ersi and its licensors.

The net widening trend in southern California is much stronger than northern California, and this trend can be approximated using linear regression ($\pm 95\%$ c.i.) to be $+56,000 \pm 17,000$ m²/yr, which is equivalent to $+2.3 \pm 0.7$ million m² of beach area formation over the 41-yr measurement record. These rates of change are equivalent to an average beach widening of $7.2 \pm 2.2$ m over the 41-yr observations (or $0.18 \pm 0.05$ m/yr) integrated across the 320 km of southern California beach length evaluated herein (Supplementary Fig. S1). Although there is strong year-to-year variability in the beach area of southern California, the computed linear trends are not sensitive to the initial year used for the time series (Supplementary Table S1). Using back-beach locations measured at the toes of cliffs, dunes or urban development (refer to Methods), we calculate that roughly 23 million m² of total beach area existed in southern California at the beginning of the satellite-derived shoreline records in 1984. This implies that there was approximately 10% expansion in total beach area within southern California during the 41-yr measurement interval. However, as detailed in the subsequent sections of this paper the rates of shoreline change over scales of littoral cells and beach segments vary considerably—including both positive and negative (i.e., widening and narrowing) trends—and these changes are related to the physical settings and historical context of these sites.

### Beach segment change in Southern California
How is net beach widening across southern California reconciled with reports of beach loss and storm damage to infrastructure across the region[27,37]? And, why are southern California's beaches widening at a rate greater than northern California considering the latter experienced a massive introduction of new sediment[49] during the late 1980s and 1990s? To address these questions, we examine the beach segment scale and document high variability in shoreline change rates from linear regression, which range between −1.7 and 2.0 m/yr over the 1984–2024 records (Fig. 2). Narrowing (red) and widening (blue) beach segments are distributed across the entire extent of the southern California coast, and narrowing and widening sections generally co-occur within the eight primary littoral cells (Fig. 2). Overall, we find that beach segments with significant widening trends ($p < 0.05$) represent 155 km of the 320 km total shoreline length (49%) assessed in our analyses of southern California, while significant narrowing is found on beach segments that combine to represent 100 km (31%) of the shoreline. No trend ($p > 0.05$) was measured on beach segments totaling 45 km (20%) of the shoreline.

The association between beach widths (measured from the toe of cliffs, dunes or urban development; refer to Methods) and shoreline trends is shown in Fig. 2b, illustrating the general pattern of how the widest beaches of the region have been exhibiting the greatest widening shoreline trends during 1984–2024. The most rapidly widening beaches correspond to the McGrath State Beach, Huntington Beach, and Venice Beach segments, which are labeled i-iii in Fig. 2a respectively, and these three beach segments were all roughly 200 m wide at the end of the 1984–2024 observations (Fig. 2b). Thus, the beach-segment-scale observations provide evidence that there is great

diversity in shoreline trends across southern California. Yet, these results also provide evidence that beach widening is more common than narrowing, that the broadest beach segments are widening generally at the most rapid rates, and that the net trend across southern California is not erosional as suggested in other studies with more uncertain data that capture only several historical shoreline postions[37].

## Southern California littoral cells

Given the high spatial variability in shoreline trends, we now examine shoreline change across the eight primary littoral cells of the southern California coast. For each littoral cell, we integrate all beach segment measurements to derive measurements of total beach area relative to the initial shoreline positions of our records (Fig. 3, lefthand column). We also provide these littoral-cell results in units of mean shoreline position integrated across each littoral cell (Fig. 3, righthand column).

Six of the eight littoral cells exhibit significant ($p < 0.05$) increasing trends in beach area, five of which are highly significant ($p < 0.001$; Fig. 3). The largest increases in beach area are found in the Santa Barbara, Santa Monica and San Pedro littoral cells. Independently, these account for 0.48-to-0.66 million $m^2$ of net beach growth, as determined from linear regression slopes, over the 1984–2024 records (Fig. 3a,c,d). If considered together, these three cells contribute to an average beach growth rate of $44,000 \pm 9000$ $m^2$/yr ($\pm 95\%$ c.i.), which would account for 1.8 million $m^2$ of new beach area over the 1984–2024 records. Of these littoral cells, the San Pedro has widened most persistently, resulting in a beach growth trend of $16,000 \pm 2000$ $m^2$/yr, which is equivalent to an average beach widening rate of $0.61 \pm 0.07$ m/yr (Fig. 3d). This trend is equivalent to $25 \pm 3$ m of average widening across the San Pedro littoral cell during the 1984-2024 records. The Zuma littoral cell, in contrast, exhibits highly significant narrowing that results in the loss of approximately −0.3 million $m^2$ of beach area; equivalent to a mean integrated shoreline change rate of $−0.18 \pm 0.07$ m/yr (Fig. 3b). The net change of beaches is more modest in the remaining four littoral cells: Laguna, Oceanside, Mission Bay and Silver Strand (Fig. 3e-h). Although the Laguna and Mission Bay littoral cells exhibit highly significant widening trends, these cells encompass relatively short lengths of shoreline and therefore are not major contributors to the overall net beach area expansion across southern California (Fig. 3e,g). In addition to trends, all littoral cells exhibit year-to-year variations in beach area, including both multi-year intervals of beach widening (wide intervals generally occur during 1999–2009 and 2012–2015) and years with punctuated narrowing (such as 1991, 1998, 2010 and 2016; Fig. 3).

The combination of the beach-segment-scale and littoral-scale shoreline trends provides evidence that most of the net widening observed across southern California beaches can be attributed to high rates of widening from a limited number of beach segments within a few of the littoral cells. For example, the most rapidly widening beach segment of southern California is McGrath State Beach in the Santa Barbara littoral cell (labeled 'i' in Fig. 2a), where the mean widening trend of 1.97 m/yr ($r^2_{adj} = 0.92$; Supplementary Fig. S2) for this 6.4-km beach segment explains 0.52 million $m^2$ of beach growth over the measurement records. This amount of beach growth is equivalent to ~80% of the total expansion in beach area integrated across the Santa Barbara littoral cell (~0.66 million $m^2$; Fig. 3a), even though the McGrath State Beach segment represents only 7.0% of the integrated beach length sampled of this cell. Expanding these analyses, we find that the four beach segments with the highest rates of beach widening represent less than 22 km of shoreline, or only 6.8% of the cumulative shoreline length of the study area; however, when considered in unison these segments can account for 65% of the total net widening trend across southern California (i.e., 1.5 of the 2.3 million $m^2$ of new beach area). These four rapidly widening beach segments have steady rates of change over time (all linear $r^2_{adj} > 0.9$; Supplementary Figs. S2-S4),

which implies that the causes of widening for these segments have been regular and persistent over the past 4 decades.

Although the rapidly widening beach segments contribute substantially to the net beach area growth in southern California, narrowing beach segments also exist throughout southern California resulting in erosional impacts to property and natural resources[45,47,50]. Examples include most of the Zuma littoral cell and the northernmost section of the Oceanside littoral cell (Fig. 2a), both of which exhibit narrowing beach segments that have caused erosion challenges to these coastal communities[51,52].

## Drivers of shoreline trends

Shoreline change can be attributed to one or more factors including new sediment supply from river discharge and bluff erosion[49,53], sediment management activities (e.g., beach nourishment or sediment bypassing of harbors)[7], net cross-shore exchange of sediment from time-varying wave energy[12,31,46], longshore sediment transport[26], and/or losses of beach sediment to offshore sediment pathways such as submarine canyons[25]. Here we examine the role of these factors across southern California littoral cells to evaluate the causes of the observed shoreline changes.

We will first examine the effects of sediment budgets, owing to observations that sediment inputs from watersheds and human activities such as beach nourishment can dramatically alter beach widths in California[32,49,54,55]. Our approach is to examine littoral cells with the greatest rates of beach widening, the Santa Barbara, Santa Monica and San Pedro littoral cells, by accounting for river inputs, nourishment projects, harbor bypassing and the net littoral volume change estimated from observed trends in shoreline position (refer to Methods; Table 1; Fig. 4). The available data are incomplete for these littoral cells, especially with respect to sediment introduced by river discharge or cliff erosion and lost offshore through submarine canyons. Thus, we focus on regions with constrainable sediment budgets that surround the most rapidly widening beach segments (denoted with dashed lines in Fig. 4a-c) to evaluate if shoreline changes may be related to measurable changes in littoral sediment volumes.

Summing sediment inputs and outputs reveals net gains in sediment volume ($+0.16$ to $+0.42$ $Mm^3$/yr) for all three sections of coast evaluated for sediment budgets, even under the range of conditions resulting from 2-$\sigma$ uncertainties (Table 1). This result indicates that these coastal sections have been gaining millions of cubic meters of littoral sediment ($+6.6$ to $+17$ $Mm^3$) over the 1984–2024 beach change records assessed here. The source of this sediment surplus varies across these sites: river supply and harbor sediment bypassing in the eastern portion of the Santa Barbara littoral cell, littoral drift in the Santa Monica littoral cell, and a combination of river supply (delivered by channel maintenance) and nourishment in the San Pedro littoral cell (Fig. 4a-c).

We compare these results of gaining sediment budgets with estimates of the measured volumetric changes across these beaches derived from our shoreline change measurements. Without repeated topographic and bathymetric survey data over the 1984–2024 measurements, we are limited to approximating these volumetric changes with simple assumptions about the beach profile shape, the integrated thickness of the active beach profile, and total uncertainty related to a worst-case scenario (refer to Methods). Using these approximations, we estimate that $+0.14$ to $+0.21$ $Mm^3$/yr ($\pm 50\%$ uncertainty) of new littoral sediment was incorporated into these beach sections from the shoreline change measurements (Table 1). Comparisons of the sediment budget and beach volume estimates provide evidence that the persistently widening beach segments of southern California are all areas with persistent increases in the coastal sediment volumes (Table 1). However, we should not expect the volumetric change estimates to compare perfectly owing to a number of factors, including

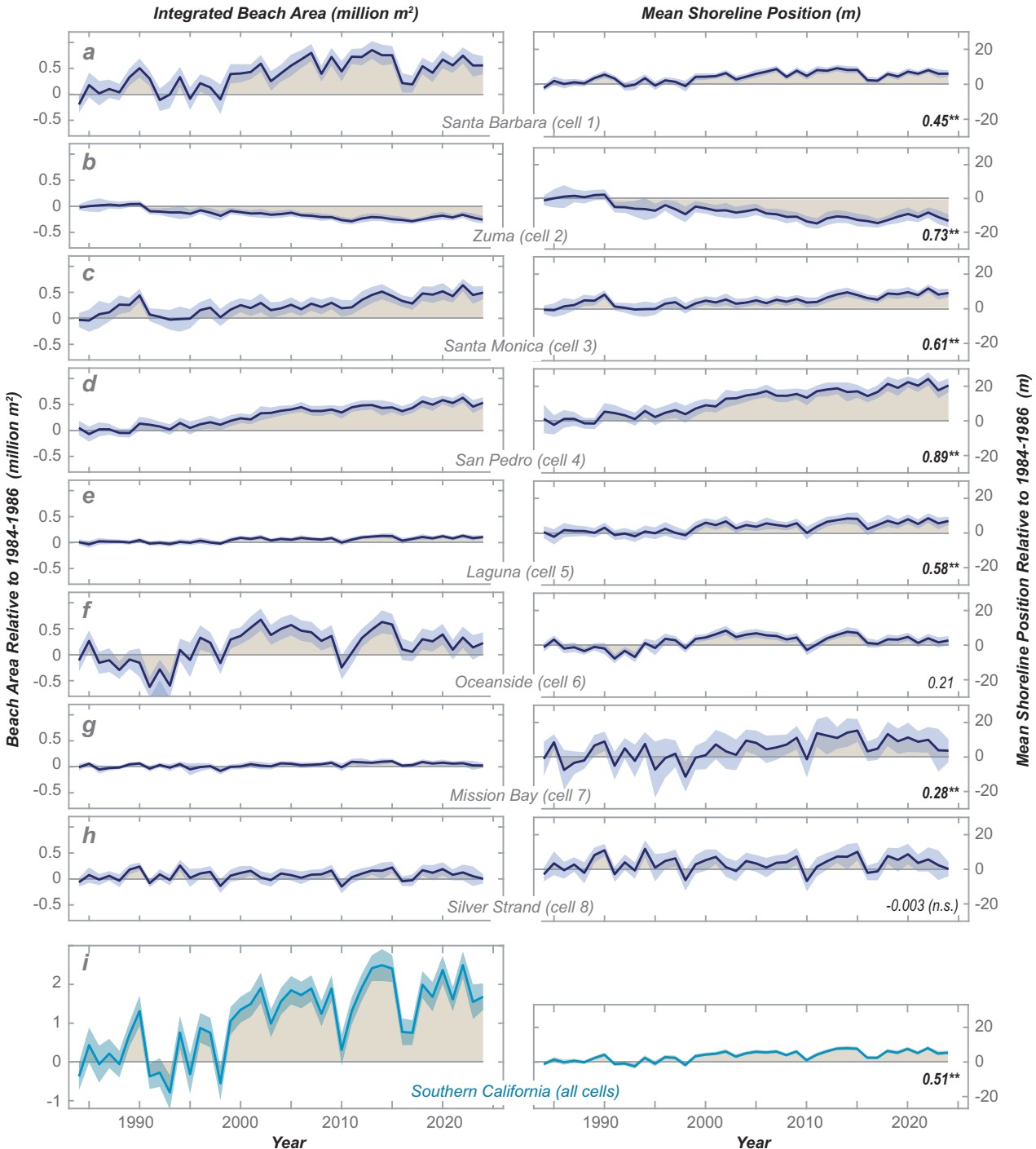

**Fig. 3 | Annual beach area and mean shoreline positions for the eight littoral cells of southern California.** Annual integrated beach area (lefthand column) and mean shoreline position (righthand column) relative to the average value of the first three years of the measurements (1984–1986) for (**a**–**h**) the eight primary littoral cells of southern California (refer to Fig. 1a for locations) and (**i**) the integrated southern California coast. Annual values (lines) include 2-σ total uncertainty (blue shading); tan shading provides total overall change from the mean 1984–1986 values. On the righthand side, the adjusted correlation coefficients ($r^2_{adj}$) are provided from linear regression, which is denoted as highly significant ($p < 0.001$) with bold text and stars, significant ($0.001 < p < 0.05$) without notation, or not significant ($p > 0.05$) with text (n.s.).

but not limited to the possibility that the mean beach profiles may have changed over time[53,54], the potential for offshore sediment losses from the littoral cells[53,54,56], variations in sediment bulk density and grain-size distributions[57], and the potential for documentation errors and omissions during nourishment projects[58]. In fact, the three coastal sections considered in detail here have widened at rates that are lower than expected from the sediment budgets, which may be indicative of offshore losses or submarine storage of sediment that was not captured by shoreline position measurements. Nevertheless, these budgets show that the supplies of sediment from rivers, littoral drift, and nourishment projects scale with the measured rates of shoreline change, providing evidence that seaward beach expansion has been occurring where millions of cubic meters of sediment have been accumulating along the coast (Table 1).

**Table 1 | 1984-2025 sediment budgets for sections of littoral cells with net beach widening**

| Littoral sediment budget element | Santa Barbara littoral cell at Santa Clara River mouth and McGrath State Beach segments | Santa Monica littoral cell at Will Rogers Beach, Santa Monica Beach and Venice Beach segments | San Pedro littoral cell at Huntington Beach and Newport Beach segments |
|---|---|---|---|
| (1) Upcoast littoral drift (Mm³/yr) | - | +0.22 ± 0.14 | - |
| (2) River inputs (Mm³/yr) | +0.60 ± 0.20 | - | +0.06 ± 0.02 |
| (3) Beach nourishment (Mm³/yr) | - | - | +0.36 ± 0.11 |
| (4) Net change from harbor bypass & dredging (Mm³/yr) | −0.29 ± 0.09 | −0.06 ± 0.02 | - |
| Total sediment budget (sum of 1–4) (Mm³/yr) | +0.31 ± 0.22 | +0.16 ± 0.14 | +0.42 ± 0.11 |
| Littoral volume change from remote sensing (Mm³/yr) | +0.15 ± 0.08 | +0.14 ± 0.07 | +0.21 ± 0.11 |

Sediment budget elements are derived from published records that best coincide with the 1984–2024 shoreline change records for each section of shoreline (refer to Methods). Locations of these three shoreline sections are provided in Fig. 4. The change in littoral sediment volume is calculated from the product of the remotely sensed shoreline trends, the beach segment lengths and an estimate of the littoral thickness, which is defined to be the height between the upper beach surface and the depth of closure. Uncertainty provided at the 2-σ level.

Attention now turns to longshore sediment transport, the primary mechanism by which sediment is distributed along the length of littoral cells and a fundamental characteristic of the littoral cells of southern California[9,25]. We acquired estimates of wave height, period and direction at the 10 m isobath every 100 m along the coast from the CDIP-MOP nowcasting system[59], which were used to estimate the direction and magnitude of sediment transport using the CERC equation (refer to Methods). The predominant transport directions are presented in Figs. 4a-c and 5a-b, with arrows showing segments of the coastline where the longshore transport is either unidirectional or bidirectional. Bidirectional transport stems primarily from seasonal changes in the prevailing wave direction, with more westerly wave energy prominent in winter and spring and more southern wave energy prominent in summer and fall, with an overall effect of diffusive sediment transport[52]. Moreover, whether any section of the coast experiences unidirectional or bidirectional transport further depends on the orientation of the coast, sheltering of wave action by offshore islands, and coastal bathymetry[52].

Littoral transport directions reveal that many of the rapidly widening beaches are areas of net sediment convergence along the littoral cells. For example, convergence in the littoral sediment transport is computed for the Huntington Beach segments of the San Pedro littoral cell, and this region of convergence corresponds with the most rapid rates of beach widening (Fig. 4c). Within the Santa Barbara and Santa Monica littoral cells, we do not compute littoral sediment transport convergence in the regions where the most rapid beach widening occurs (Fig. 4a, b). However, convergence of southward directed sediment transport in these cells is supported with sediment budget considerations. For example, convergence of the southward moving littoral sediment is enhanced at the McGrath State Beach segment of the Santa Barbara littoral cell owing to harbor bypass operations that do not keep up with sediment inputs (Fig. 4a). Similarly, convergence of southward transport of sediment in the Santa Monica littoral cell is likely caused by interruption of the littoral drift by coastal structures including Marina del Rey and the Santa Monica and Venice Breakwaters, which are not adequately captured in our longshore transport computations (Fig. 4b, e). In the end, the littoral sediment flux estimates combined with sediment budgets provide evidence that these widening beach segments have both net gains in sediment volume and net sediment transport convergence, resulting in the addition of millions of cubic meters of sand to these widening segments of beach (Fig. 4).

Lastly, we consider the effects of cross-shore exchange of littoral sediment on the beach trends reported above, because changes in incident wave energy over time can result in time-dependent changes to the shoreline position[4,12,31,46]. Much of the west coast of North America experiences shoreline erosion during intervals of elevated incident wave energy, such as in 1998, 2010 and 2016, with shoreline recovery occurring over post-storm years[45–48,50,60]. To examine whether the trends in beach area across the eight littoral cells of southern California can be explained by time-dependent cross-shore sediment exchange at the annual scale, we utilize mean annual wave power hindcasts from the ERA5 models[61] because these models are the only complete hindcast covering our 1984–2024 measurement records (refer to Methods).

We find evidence that wave power induces significant effects on shoreline positions and integrated beach areas in southern California. For example, there are significant correlations between the annual records of wave power and beach area for most littoral cells of southern California (Table 2). These wave power correlations are strong for the littoral cells with weak to moderate temporal trends and weak for the cells with strong temporal trends. Overall, wave power explains between 0 and 51% (average = 31%) of the year-to-year variations in beach area across these eight littoral cells (Table 2). These correlation results can be compared to the time-dependent trends that explain between 0 and 89% (average = 47%) of the variations in the littoral cell beach area (Table 2). The wave power correlations all have inverse relationships, revealing that years with larger waves generally coincide with smaller beach areas whereas smaller waves coincide with broader beaches, consistent with wave-related cross-shore exchanges of sediment.

Because both time and waves exhibit significant correlations with beach area, we examine the combined effects of these factors using multivariable linear regression, which reveals that both variables generally contribute significantly to explaining beach area across the littoral cells (denoted with [time + waves] in Table 2). For example, over the broader region of southern California the $r^2_{adj}$ values increase from 0.52–0.56 for the single-variable linear regressions to 0.81 for the multivariable regression, and both variables (time and waves) are found to be significant at $p < 0.05$ (Table 2). At the littoral-cell scale, the combination of time and waves explains between 30 and 94% (mean = 65%) of the beach area variability, and these correlations are substantial improvements over the single-variable correlations for most littoral cells (Table 2). Thus, we find that the combination temporal trends and time-varying wave power contribute to explain much of the changes to beach area across southern California.

Yet, we also find that wave power from the ERA5 hindcasts exhibits a weak decreasing trend over the 1984-2024 records ($r^2_{adj} = 0.083$; $p < 0.05$; Supplementary Fig. S5), so we evaluate whether these trends may explain a portion of the changes in beach area. The effects of waves on overall beach trends can be evaluated by comparing the

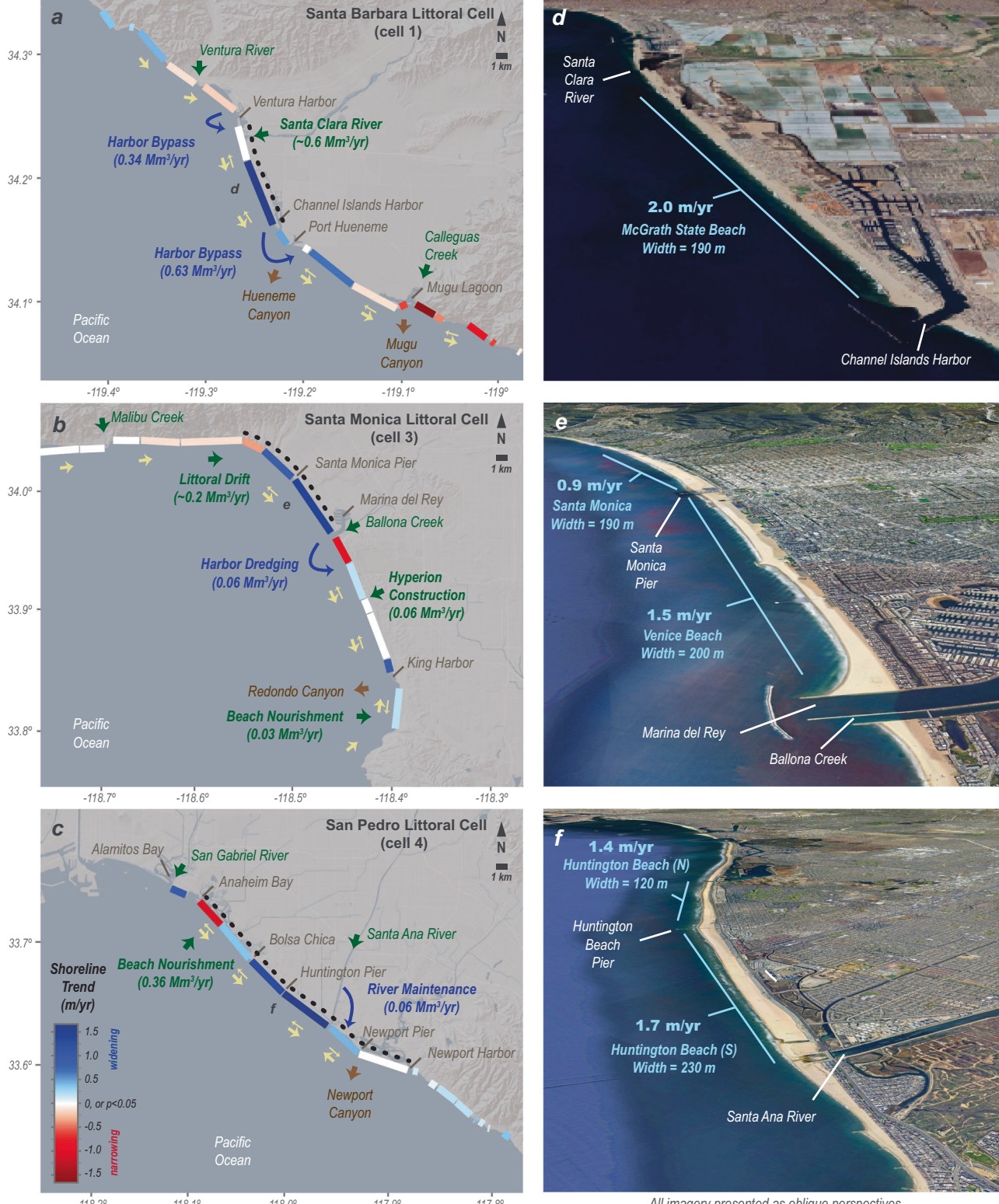

**Fig. 4 | Beach changes and littoral sediment budgets for widening beaches of California. a–c** Site maps of three sections of the southern California coast with the highest rates of beach widening showing mean annual shoreline trends and sediment budget elements. Arrows show sediment transport including sediment sources (green), littoral transport (yellow), sediment management (blue), and sediment sinks (brown). The geographic extent of the sediment budget calculations was limited to portions of the littoral cells (dashed lines), and the mean annual sediment budgets are summarized in Table 1. Littoral transport provided by reach-averaged Peclet numbers, showing advection-dominant transport (single arrows) and diffusive-dominant transport (double arrows). **d–f** Oblique perspectives of satellite imagery for beach segments with the fastest rates of widening. Base maps in (**a–c**) provided by Ersi and its licensors; data and imagery in (**d–f**) provided by Google, USGS, CSUMB SFML, CA OPC, SIO, NOAA, U.S Navy, NGA, GEBCO and ©2025 Airbus.

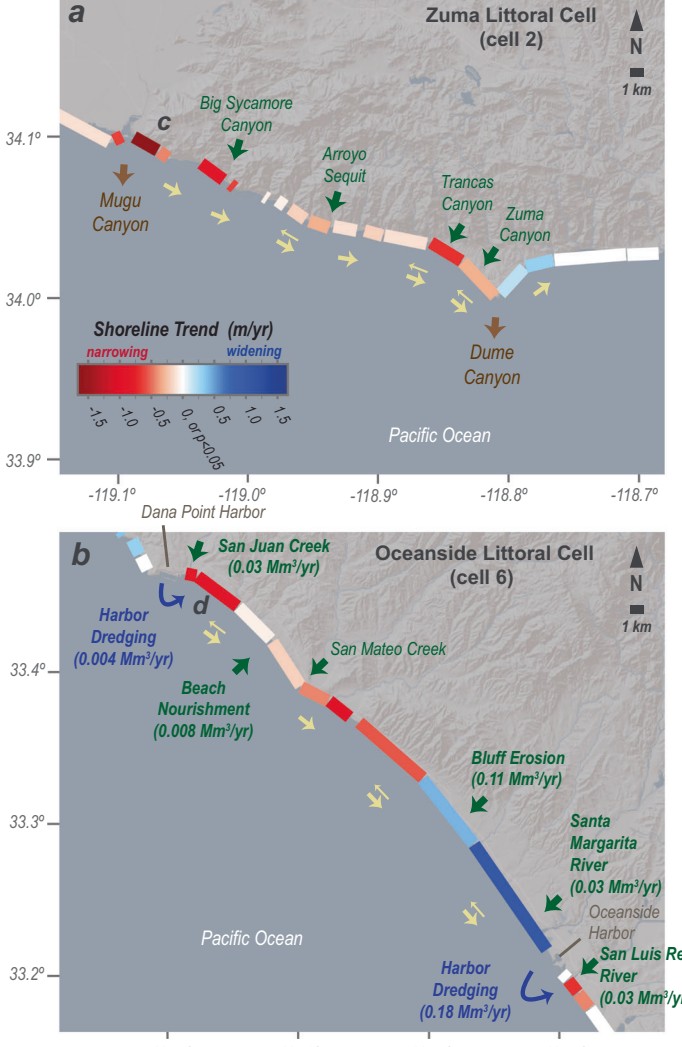

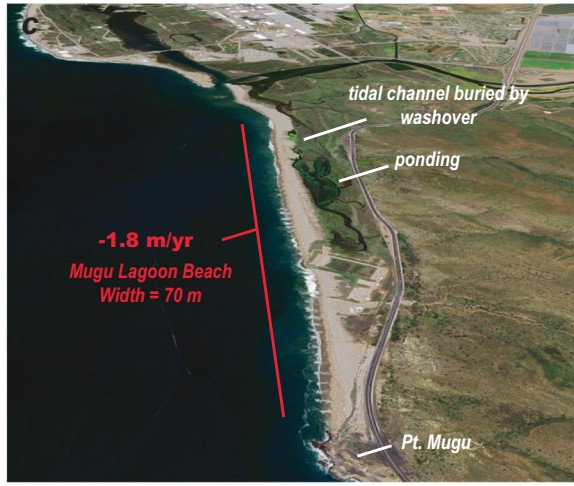

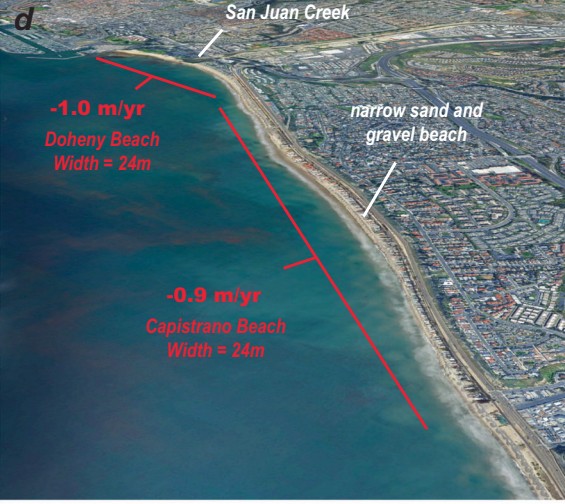

*All imagery presented as oblique perspectives.*

**Fig. 5 | Beach changes and littoral sediment budgets for narrowing beaches of California. a, b** Site maps of two sections of the southern California coast with high rates of beach narrowing showing mean annual shoreline trends and sediment budget elements. Arrows show sediment transport including sediment sources (green), littoral transport (yellow), sediment management (blue), and sediment sinks (brown). Elements of the sediment budgets are provided in Table S1 of the Supplemental Information. Littoral transport provided by reach-averaged Peclet numbers, showing advection-dominant transport (single arrows) and diffusive-dominant transport (double arrows). **c, d** Oblique perspectives of satellite imagery for beach segments with the most rapid rates of narrowing. Base maps in (**a, b**) provided by Ersi and its licensors; data and imagery in (**c, d**) provided by Google, CSUMB SFML, CA OPC, SIO, U.S. Navy, NGA and ©2025 Airbus.

single-variable temporal trends with the trends computed from multivariable regression. When the effects of waves are included, the temporal trends are found to be slightly lower than the trends from linear regression, although the two temporal trend estimates are not different within 2-σ uncertainty for any of the littoral cells (Table 2). Combined, this suggests that wave-related cross-shore exchanges of sediment are important in year-to-year variations in beach area, causing shoreline erosion during years with large waves and beach recovery during years with small waves, consistent with field and modeling results[31,45,46]. However, cross-shore exchanges of sediment cannot explain the significant temporal trends exhibited in seven of the eight littoral cells of southern California, which combined with the sediment budget and longshore transport results highlighted above indicates that the shoreline response can be largely attributed to substantial increases in the littoral sediment volumes.

## Discussion
Accurate characterization of shoreline trends shown herein has been made possible by newly developed satellite-derived-shoreline data and

signal-processing techniques[44], which if applied more broadly will assist in revolutionizing assessments of beaches and their changes throughout the world. Using these new tools, we have been able to characterize regional differences [100–1000 km] of change across the California coast and examine the sources and causes of these differences at littoral [10–100 km] and beach segment [0.5–10 km] scales (Figs. 1–3). Using an annual basis for evaluating beaches, these techniques reduce shoreline position uncertainties by almost an order of magnitude from values of raw satellite-derived shoreline data[62], and they enable the production of shoreline records that span multiple decades.

These results provide the somewhat counterintuitive finding that the most heavily populated, dammed, urbanized, and impacted coastal region of California has experienced recent net growth in its beaches, which contrasts with prior reporting that the southern California region has transitioned to a state of chronic beach erosion[27,37]. Assessments of sediment discharge in southern California's watersheds have suggested that sediment removal from dams and debris basins throughout the region has reduced sand discharge to the coast

**Table 2 | Regression statistics for time-dependent beach areas of the littoral cells of southern California**

| | Santa Barbara (cell 1) | Zuma (cell 2) | Santa Monica (cell 3) | San Pedro (cell 4) | Laguna (cell 5) | Oceanside (cell 6) | Mission Bay (cell 7) | Silver Strand (cell 8) | All southern California |
|---|---|---|---|---|---|---|---|---|---|
| $r^2_{adj}$ [time] | 0.45** | 0.73** | 0.61** | 0.89** | 0.58** | 0.21 | 0.28** | -0.003 (n.s.) | 0.52** |
| $r^2_{adj}$ [waves] | 0.46** | <0.01 (n.s.) | 0.19 | 0.20 | 0.37** | 0.41** | 0.51** | 0.30** | 0.56** |
| $r^2_{adj}$ [time+waves] | 0.71** tw | 0.72** t | 0.64** tw | 0.94** tw | 0.74** tw | 0.50** tw | 0.65** tw | 0.30** w | 0.81** tw |
| Temporal trend from [time] regression ($10^3$ m²/yr) | 16.2±5.6 | -6.7±1.3 | 11.6±2.7 | 15.9±1.7 | 3.1±0.8 | 12.8±7.5 | 2.1±1.0 | 1.2±2.6 | 56.1±17.4 |
| Temporal trend from [time+waves] regression ($10^3$ m²/yr) | 13.0±4.5 | -7.1±1.4 | 10.5±3.0 | 15.2±1.5 | 2.6±0.7 | 9.1±6.6 | 1.6±0.8 | 0.2±2.4 | 45.0±12.0 |
| Ratio of temporal trends [time+waves]:[time] | 0.80 | 1.06 | 0.91 | 0.95 | 0.84 | 0.71 | 0.76 | 0.17 | 0.80 |

Adjusted correlation coefficients ($r^2_{adj}$) are provided for linear regressions between mean annual beach area and year [time], mean annual wave power [waves] and multiple variable linear regression with year and wave power [time+waves]. Regression coefficients are denoted as highly significant ($p < 0.001$) with stars (**), significant ($0.001 < p < 0.05$) without notation, or not significant ($p > 0.05$) with text (n.s.). The multiple variable regressions are denoted with the variables that were found to be significantly different from zero using 95% confidence intervals (t = time, w = waves). The temporal trends of the [time] and [time+waves] regressions are provided with uncertainty at the 2-σ level.

by ~50%[24,28]. Because the littoral cells of southern California are highly dependent on river sand sources to replenish beach sand lost to the deep sea through submarine canyons, it is widely expected that reductions of river sand discharge would result in shoreline retreat and beach narrowing[24,27,28,37]. However, our findings that southern California's beaches are experiencing net widening because of a surplus of littoral sand and convergence of littoral sediment transport contradict these expectations of beach erosion, and our findings should provide caution about applying simple assumptions about the impacts of dams and land use to coastal change trends. Because of this contradiction, we recommend that evaluations of watershed modifications, such as dams, on shoreline trends should include rigorously measured and evaluated coastal change assessments, especially at beach-segment, littoral-cell and regional scales. Although global assessments of the human impacts of sediment discharge to the sea provide guidance about the general causes and scales of river sediment disruption[11,63], extrapolation of these global-scale findings to regional and littoral-cell scales may result in incorrect notions about the trends and causes of coastal change, as shown above for the southern California region[37].

Our findings about beach status and trends provide new opportunities for improved coastal management and sustainability. For example, our measurements provide strong evidence that the overall input of sand to southern California beaches is not the underlying problem behind localized beach erosion in the region. Rather, we find that the total input of sand is adequate to induce net widening of beaches across the region, but that littoral sand is simply unable to move to beach locations where erosion problems are greatest, owing to a combination of blocking of southward sand transport by harbors and coastal structures and natural patterns of littoral sediment transport in the region. Rates of beach widening are persistent and greatest at the beach locations that trap sediment on northern sides of infrastructure or at littoral-cell convergence zones (Fig. 4). Although the seasonal to interannual erosional impacts to beaches by cross-shore sediment transport during large waves events can be exceptional[45–47,60], these storm-related erosional impacts during years such as 2010 and 2016 are generally short-term and recoverable (Figs. 1 and 3). In contrast, sand trapping impacts along the coast related to littoral drift and sediment convergence are chronic and not easily recoverable. Evidence for these chronic patterns exist in the persistent trends exhibited at the beach segment and littoral cell measurements that are highlighted in Figs. 2–5.

Combined, these findings indicate that human activities have strongly influenced the rates, locations and amounts of beach change in southern California during our measurement records (1984–2024). Although earlier mid-20th century beach expansion has been well documented for southern California as a result of construction-related additions of sediment to the littoral cells[26,27,32], here we have shown that similar kinds of human impacts have continued through the late 20th and early 21st centuries. Perhaps these human impacts are easiest to observe in the San Pedro littoral cell (Fig. 3d), where beach nourishment using sandy sediment from the inner continental shelf is several times larger than river sediment supplies, resulting in massive beach expansion centered near Huntington Beach (Fig. 4c). As such, the widening of beach segments in this littoral cell is directly related to both the human nourishment activities at the northwestern extent and the accumulation of longshore transport in the middle of this littoral cell. Overall, these findings are consistent with other coastal regions with persistent nourishment and human-built coastal structures[7,9,64].

Although there is net beach widening across southern California, this region is also challenged by local effects of sediment deficit. Beach erosion remains a persistent and challenging condition for many southern California communities with damaging impacts to property, infrastructure and resources[26,51]. Indeed, our results show that 30% of the shoreline length of the southern California region experienced significant beach narrowing during 1984–2024 (Fig. 2). The Zuma

littoral cell and the northernmost portion of the Oceanside littoral cell have some of southern California's most widespread shoreline erosion, and both areas have beach segments that have been eroding at rates exceeding 1 m/yr during our measurement records, which has critically narrowed beaches and caused erosion and wave-related damage to public and private property[46,51,65] (Fig. 5; Supplementary Figs. S6–S7).

Globally, there has been increasing documentation of coastlines degraded by sand shortages, brought on my mining, land development and dam construction activities[11,66]. However, our work suggests that in southern California, the supplies of sediment have been adequate at the regional scale, and the primary challenge—and opportunity—stems from an uneven distribution with excessive sediment volumes in zones of convergence (e.g., in beaches updrift of coastal structures, within harbors, or in inland reservoirs formed by dams)[24,26] and inadequate sediment volumes in zones of divergence (i.e., erosion hot spots or beaches downdrift of coastal structures interrupting longshore transport)[26,51]. Overcoming this challenge could result in more equitable, widespread, and natural use of littoral sediment resources and enhanced coastal sustainability, and to a degree, this work is already underway. At several harbor outlets along the southern California coast including Santa Barbara, Ventura, and Channel Islands/ Port Hueneme, sediment bypass systems are in place which move sediment past barriers, and the routine maintenance dredging of southern California harbors and channels can move sediment from areas of convergence to eroding beaches[26]. Additionally, dam removal projects are being planned that may help with erosion hot spots to the extent that eroded beaches are positioned within their littoral influence[67,68]. Yet, there remains many recognized barriers to refining and improving sediment management activities, including financial costs, regulatory processes, institutional inertia, political boundaries, and human resistance to change[69]. In light of these challenges, others have pointed toward systemic changes that could make beneficial sediment management a more accessible option for southern California and elsewhere, resulting in better coastal resilience and sustainability[69,70].

In conclusion, the introduction of decades of spatiotemporally dense satellite-derived shoreline data with new analysis techniques to reduce error are to our knowledge the only means to reveal spatial variability in the trends and human impacts to beaches at the regional-to-beach-segment scales reported herein. As such, the continued use of these remotely sensed shoreline data with advanced signal processes techniques, such as those developed herein, could provide important information for understanding and managing coastal systems now and into the future.

## Methods

For the California study area, there have been regular field-based measurements of beach morphodynamics at a limited number of sites, which if combined represent only 2.5% of the study area's shoreline[36,46,71]. Because of the limited spatial extent of these data and the high variability of shoreline behavior across the study area[43,49,72], we have taken a remote sensing and signal processing approach to capture beach changes along the 1700 km coastline over a four-decade interval of time. To implement this strategy, we use the latest CoastSat satellite-derived shoreline position measurements[41,42] and a time-series signal processing approach[44] based on the decomposition analyses of Warrick et al.[43] to reduce measurement errors. These remote sensing results are combined with assessments of wave-related shoreline change, longshore transport, and littoral sediment budgets to assess the patterns and causes of coastal change.

### Shoreline measurements

Shoreline change analyses utilized measurements from the CoastSat technique, which combines Landsat and Sentinel-2 satellite imagery, machine learning-enhanced radiometric analyses, and tidal models to estimate time-series of shoreline positions at regularly spaced transects[41,62]. For each available and cloud-free satellite image, waterlines are mapped using radiometric properties of the infrared and optical bands, segmentation of the scene into water, whitewater, sand, and other land covers, an Otsu radiometric threshold for each image to delineate the waterline based on segmentation boundaries, and a global tidal model and local estimate of beach slope to correct the waterline measurements to a mean-sea-level shoreline position[41,73]. Shoreline data for the California beach segments in our study were derived from a study of shoreline change across the Pacific Ocean with beach transect spacing at 100 m[43,50]. We utilized the most recent release of these shoreline data at the time of our data analyses (version 1.5), which included updates in transect beach slopes and the length of record (1984–2024)[42].

CoastSat data were used to compute mean annual shoreline positions and beach area changes for distinct segments of the California littoral cells. Signal processing techniques were used to produce regular interval time-series of shoreline positions, reduce the uncertainty in satellite-derived shoreline positions, and characterize the patterns, behaviors, and causes of beach change. The analysis approach was based on the newly developed time-series decomposition analyses of Warrick et al.[43], which utilizes the seasonal-trend decomposition with locally estimated scatterplot smoothing (LOESS) (STL) technique and is summarized for two example beach segments in Supplementary Fig. S8. The code used to conduct this analysis is available from Warrick[44].

Preprocessing of the raw CoastSat shoreline position data included removal of extreme outliers, if they occurred, using a Hampel filter run across 5% subsets of data from each transect and removing all shoreline position values that were greater than 3 standard deviations from the local mean. Monthly mean shoreline positions were then calculated from the preprocessed values for each transect over the range of January 1984 through December 2024. These monthly shoreline position records were demeaned by subtracting average shoreline positions of the entire time series values on a transect-by-transect basis.

The demeaned monthly shoreline positions were then integrated into 329 beach segments of the California coast, 129 of which are in southern California. Each beach segment represents a unique geographical setting along the study area, and the geographic bounds for these beach segments were defined by the locations of features including headlands, breakwaters and groins, harbors, river mouths, piers, changes in shoreline orientation, or changes in shoreline change trends. We excluded all transect segments that included shoreline position irregularities caused by land-cast shadows, offshore rocks, coastal structures such as piers, and the opening and closing of river mouth inlets. These removed transects are consistent with those removed by the seasonal analyses of Warrick et al.[43]. Details of the geographic extent of these beach segments, including source CoastSat transect numbers and geographic bounds are tabulated in the data publication associated with this paper[74]. For each beach segment, mean monthly shoreline positions were calculated from the demeaned monthly shoreline positions for all transects within the segment. Examples of monthly shoreline positions from two beach segments are provided in Supplementary Fig. S8a.

For each beach segment, the annual mean shoreline positions were calculated using a decomposition technique that removed seasonal effects in the shoreline position data. This 'deseasonalizing' of the monthly records was necessary owing to data gaps in the monthly records, especially during 1984–1999 when entire seasons of the records did not have shoreline measurements (Supplementary Fig. S9). Without corrections for the typical seasonality of each beach segment, the annual mean values or other averaging techniques could be biased if the shoreline data were sourced primarily from the portion of the year during which the beach was wider or narrower than normal.

The deseasonalization process followed the seasonal, trend decomposition with LOESS (STL) techniques proposed by Cleveland et al.[75] and implemented by Warrick et al.[43] to measure shoreline seasonal cycles for the California coast. First, a LOESS fit through each monthly shoreline time series was computed to determine the time-dependent trend of each beach segment (blue lines in Supplementary Fig. S8a). This LOESS low-pass fit used the parameterization of Warrick et al.,[43] which included a single-pass of the standard second order polynomial fit with a tri-cube weight function and a LOESS bandwidth (or 'smoothing factor') of 4 years. The LOESS low-pass was subtracted from each monthly shoreline time-series, resulting in detrended monthly shoreline positions for each beach segment. The regular seasonal cycles of the shoreline within each beach segment were then calculated by averaging all of the detrended shoreline values for each month of the year, following Warrick et al.[43] To 'deseasonalize' the original monthly shoreline positions for each beach segment, these computed seasonal cycles were replicated year-after-year and subtracted from the mean monthly shoreline positions. The resulting shoreline records preserved annual to multi-annual trends while minimizing seasonal patterns as shown with example data in Supplementary Fig. S8b.

The final step in calculating mean annual shoreline positions for each beach segment was to average the deseasonalized shoreline positions over each 'wave year', which was defined as November 1 to October 31 (for example, wave year 2014 is equivalent to November 1, 2013 to October 31, 2014). The standard deviations of the available shoreline positions for each wave year were also computed from the monthly deseasonalized records, and the total 2-σ uncertainty for each wave year was calculated as the quadrature sum of 1.97 multiplied by the computed annual standard deviation values and the mean 2-σ measurement uncertainties at the annual scale, which was computed to be 5.8 meters as noted below. Lastly, for presentation and comparison purposes, the mean shoreline position of the first three years (1984–1986) of each beach segment was subtracted from the mean annual shoreline position records, so that each shoreline record tracked changes from the initial state of the time series (Supplementary Fig. S8c).

Shoreline changes within each beach segment were computed and presented in both units of change in shoreline position (m) and change in beach area (m²). Areal changes were computed by multiplying the shoreline position change by the beach length, the latter of which was calculated by the product of the number of transects in the beach segment and the transect spacing distance (100 m). Rates of change for each beach segment and integrated collections of beach segments were computed using linear regression. When integrating multiple beach segments into a regional or littoral-cell scale measurement of shoreline change, the 2-σ uncertainties for these beach sections were computed using addition in quadrature because each mean annual shoreline position at the segment scale is independent. Adjusted correlation coefficients ($r^2_{adj}$) were used to describe linear regression to provide better comparisons with results when different numbers of predictors are used. Significances of the linear regressions were assessed with p-values, including characterization of regressions as highly significant ($p < 0.001$), significant ($p < 0.05$), and not significant ($p > 0.05$).

For computations of absolute beach width and beach width change from the mean annual shoreline positions, we utilized the mapped back-beach locations derived and published for each CoastSat transect by Warrick and Buscombe.[76] These locations were based on the geospatial locations of the seaward most occurrence of a cliff toe, dune toe, vegetation line, or an urban landscape feature (e.g., road, building, riprap, seawall).

## Comparison of satellite and field-based measurements of shoreline change
The uncertainty between the satellite-derived shoreline positions and field-surveys of shoreline positions was evaluated to compute total uncertainty in the mean annual shoreline positions. For these comparisons, we used field-surveys of southern California beaches by the Coastal Processes Group at Scripps Institution of Oceanography and published in Ludka et al.[71] We combined these field data to match beach segments used to characterize the CoastSat data, which resulted in three beach segments, Cardiff Beach, Solana Beach, and southern Torrey Pines represented by field-study sections C1-C2, S1-S3, and T2-T5, respectively, from the Ludka et al.[71] data. Mean monthly shoreline values from the field surveys were available for many, but not all, of the time series interval, which were 8–15 yr long (Supplementary Fig. S10).

For each of these three beach segments, the mean annual shoreline positions were computed using the STL techniques described above for both the satellite-derived shorelines and the field-derived shorelines. A time-series comparison of the satellite- and field-derived mean annual shorelines is provided in Supplementary Fig. S11. One-to-one comparisons between these segment-averaged mean annual shorelines reveal that the RMSE between the data ranged between 1.9 and 3.4 m (average = 2.9 m; Supplementary Fig. S12). These comparisons also revealed that the linear regression slopes between the data ranged between 0.96 and 1.37 (average = 1.18). Only the linear regression slope from Cardiff Beach was significantly different than unity using 95% confidence intervals; the linear slopes for Solana Beach and Torrey Pines were not significantly different than one. Thus, there is a possibility for a subtle bias in the satellite-derived shoreline positions, such that they overestimate the total shoreline change exhibited on the beach, but there is also the possibility that this result occurred because the field-survey records do not include complete monthly records from which the comparisons were made (Supplementary Fig. S10). In the end, we utilize the mean estimate of 5.8 m for the mean 2-σ shoreline measurement error, which is several-times to an order-of-magnitude smaller than traditional shoreline mapping using historical maps and orthoimagery that have been used for previous assessments of California shoreline change[27,37].

## Evaluation of wave-related shoreline change
We evaluate the influence of waves on mean annual shoreline positions within the southern California and its eight littoral cells by comparing annual shoreline measurements and total cumulative wave power estimates. These comparisons required wave data from 1984 to 2024, which limited the available data to coarse-resolution hindcast data from the European Centre for Medium-Range Weather Forecasts Reanalysis Version 5 (ERA5)[61]. A severe limitation of the use of the ERA5 data is that they do not include wave refraction and shadowing effects caused by the complex southern California topography and bathymetry[59,77]. For these reasons, we only make comparisons at the regional and littoral-cell scales and not at the local or beach-segment scales.

Total cumulative annual wave power was computed from hourly ERA5 hindcasts of significant wave heights (H) and dominant wave period (T) for offshore sites offshore of the southern California littoral cells, and locations of these sites are shown in Supplementary Fig. S5b. Because these sites are offshore of the continental shelf, we used the deep-water equations to compute wave power (P) from each hourly wave value:

$$P = \frac{\rho g^2}{64\pi} H^2 T \tag{1}$$

where $\rho$ is the seawater density (1025 kg/m³) and g is the gravitational constant (9.81 m/s²). To generate total cumulative annual wave power ($P_T$), we integrated the computed values of P over each wave year, resulting in $P_T$ values in units of kW-hr/m. We find that hindcast estimates of wave power throughout southern California were strongly correlated to each other ($r^2 > 0.95$) and different by simple linear scaling factors (Supplementary Fig. S5a). Thus, for correlation analyses

with the beach area records (summarized below), we found negligible differences in results from the different ERA5 hindcast locations. Given these findings, we computed and used a mean wave power record from the three ERA5 records evaluated herein to represent the Southern California wave conditions. Although this technique does not incorporate the diversity in wave conditions that occur throughout the southern California region[77], we do not apply these ERA5 wave data to the local beach segment scales; rather, we apply the ERA5 wave data to bulk integrated measurements across the eight littoral cells and the entire southern California region.

The effects of $P_T$ on integrated net beach area of the southern California littoral cells were assessed with linear regressions with annual integrated beach area, and adjusted $r^2$ values for these comparisons are reported in Table 2. To evaluate the potential for beach area to be influenced by a combination of temporal trends and the effects of waves, we conducted multivariable linear regressions with both measurement year and annual $P_T$ values. We report both the adjusted $r^2$ and the temporal trend values from these regressions in Table 2.

### Evaluation of longshore transport

Longshore transport processes were evaluated across southern California's littoral cells based on the potential sediment transport ($Q$) and a derivative function, the littoral Peclet Number ($Pe$) as introduced by Kahl et al.[52]. The Peclet number provides a non-dimensional indicator of the relative strength and direction of sediment transport along the coast. We investigated the spatial evolution of the Peclet number to provide insight to the directionality and variability of longshore transport processes on a sub-littoral scale.

Potential sediment transport represents the volumetric flow rate of sand that is feasible given the available wave energy and was estimated using the CERC[9] equation:

$$Q = K\gamma\sqrt{g}H^{\frac{5}{2}}sin(2\theta_b) \qquad (2)$$

where $Q$ is the transport potential, $K$ is an empirical coefficient, $g$ is the gravitational acceleration constant, $H$ is the significant wave height, and $\theta_b$ is the relative angle to the coast of waves at breaking, computed as the difference between shore normal direction and bulk wave direction ($\theta_b = \theta_{shore} - \theta_{bulk}$). Sediment and fluid characteristics were represented by:

$$\gamma = \frac{\sqrt{2}\rho_w}{16(\rho_s - \rho_w)\phi} \qquad (3)$$

where $\rho_w$ and $\rho_s$ represent water and sediment density, respectively, and $\phi$ represents sediment porosity. Standard values were utilized for each variable as follows: $K = 0.39$, $g = 9.8$ m/s$^2$, $\rho_w = 1024$ kg/m$^3$, $\rho_s = 2650$ kg/m$^3$, and $\phi = 0.6$. Significant wave height and bulk wave direction values were extracted from CDIP-MOP hindcasts on an hourly timescale for all relevant transects; shore normal angle is provided from CDIP-MOP as a constant for each transect. Beginning at the start of the dataset availability in 2000, time series were downloaded from the CDIP THREDDS server via a data-scraping script developed in R. Potential transport rate was then estimated for each hour and averaged across the 24-year time-scale yielding a Summer (April–September), Winter (October–March), and annual mean rate (m$^3$/year). The annual mean and annual standard deviation values were used to compute the littoral Peclet Number, $Pe$, representing the ratio of advective to diffusive transport processes for each coastal transect[52]. Shoreline locations where $Pe$ is greater than or equal to 0.5 or less than or equal to −0.5 are defined to be advection dominated, with transport predominantly in the upcoast or downcoast directions, respectively. Conversely, a $Pe$ between −0.5 and 0.5 points to diffusive (i.e., bidirectional) dominated zones with transport in both

the upcoast and downcoast directions. The resulting potential transport values and Peclet numbers shown in Supplementary Fig. S13 were used to inform the location, direction, and strength of longshore sediment transport, indicated with yellow arrows across Figs. 4 and 5.

### Evaluation of beach nourishment and bypassing rates

Average annual rates of beach nourishment (Mm$^3$/year) were calculated over the full study period (1984-2024) based on nourishment volumes and project sites recorded in the American Shore and Beach Preservation Association (ASBPA) database[64] along with supplementary information from a range of open-source records summarized in Supplementary Table S2. Project sites generally corresponded to coastal municipalities, and municipal-scale nourishment rates were summed to arrive at totals for each littoral cell of the region. Additionally, nourishment volume known to be taken from river outlets or moved across harbors was classified as "bypassing." Figs. 4 and 5 use arrows to identify nourishments associated with bypassing, river maintenance, and conventional placement. Summation of these rates of sediment movement for the purpose of the beach-segment-scale sediment budgets is detailed below.

### Littoral cell sediment budgets

Sediment budgets were generated for segments of the southern California coast with the greatest beach widening and narrowing rates using available records of river sediment discharge, dredging, beach nourishment, and littoral sediment transport. The sediment budgets were derived for the same interval of time as the shoreline records (1984-2024). River sediment discharge in southern California is inherently ephemeral, and we used data from Barnard and Warrick[54] for the Santa Clara River, Warrick and Rubin[38] for the Santa Ana River, and Patsch and Griggs[26] for the creek and rivers of the Oceanside littoral cell. We used the bluff erosion rates summarized for the Oceanside littoral cell by Patsch and Griggs[26]. Harbor dredging and bypassing and beach nourishment from river maintenance, construction projects, and offshore borrow sites were derived from the mean annual rates computed from the ASBPA National Beach Nourishment Database[64] that are described in detail in the previous section. Lastly, littoral sediment transport into Santa Monica Bay was derived from the assessments of engineering reports summarized by Patch and Griggs[26].

Beach-segment-scale sediment budgets are summarized in Table 1 for three portions of the southern California coast as denoted by dashed lines in Fig. 4. We used summation of the ASBPA data provided in Supplementary Table S1 for the presentation in Table 1 and Figs. 4 and 5. For the McGrath State Beach segment in the Santa Barbara littoral cell (Fig. 4a), the net change from harbor bypassing and dredging was computed by the difference between the sediment input at Ventura Harbor and the sum of the output for Port Hueneme and Channel Islands Harbor (e.g., 0.34 − (0.14 + 0.49) = 0.29 Mm$^3$/yr). For the Santa Monica littoral cell sediment budget (Fig. 4b), the dredging of Marina del Rey (0.06 Mm$^3$/yr) was included as a sediment output from the northern beaches, but the remaining nourishment projects were not included in the sediment budget because they were conducted outside of the sediment budget study area. For the San Pedro littoral cell (Fig. 4c), the Surfside-Sunset, Huntington Beach and Bolsa Chica nourishment rates were combined (0.14 + 0.21 + 0.013 = 0.36 Mm$^3$/yr) because these projects are conducted at or near the northwestern end of the study area. The Seal Beach, Long Beach and San Pedro nourishment rates were excluded, because these beach segments are separated from the area of interest from harbor breakwaters. Lastly, the Newport Beach nourishment projects (0.06 Mm$^3$/yr) have been derived from maintenance of sand deposited in the Santa Ana River to reduce the impacts of channel sedimentation. Although there was not a formal sediment budget constructed for the Oceanside cell, we present a summation of the ASBPA data in Fig. 5 that includes dredging from Dana Point Harbor from the Capistrano Beach rate

(0.0044 Mm³/yr), total beach nourishment in the erosional segments of the northern littoral cell as the sum of San Clemente, Doheny Beach, San Clemente Pier and North Beach nourishments (0.0003 + 0.0024 + 0.0038 + 0.0010 = 0.0075 Mm³/yr), and sediment bypassing around Oceanside Harbor as the Oceanside nourishment rate (0.18 Mm³/yr).

Uncertainties in the sediment budget elements were assessed at the 2-σ level. Accumulation of uncertainty from multiple sediment budget elements was computed using the quadratic sum of the uncertainty elements, because the uncertainty elements are independent. For river sediment discharge, we used 33% uncertainty for 2-σ uncertainty, considering the sediment mass balance of Barnard and Warrick[54]. The ASBPA National Beach Nourishment Database does not include uncertainty assessments[64], so we used a conservative value of 30% for 2-σ uncertainty to incorporate the potential for incomplete or erroneous project records and the potential for changes in bulk density between the source, delivered, and native littoral sediment. Calculated rates of littoral sediment transport into northern Santa Monica Bay were reported to be 0.145, 0.206 and 0.306 Mm³/yr by Patsch and Griggs[26], so we use the mean and two times the standard deviation of 0.22 ± 0.14 Mm³/yr. This rate of sand input is consistent with sediment supplies from erosion in the watersheds and along the sea cliffs of the Santa Monica Mountains, which would feed this littoral sediment transport[78].

The sediment budgets were compared with the volumetric change estimates of beach segments. These volumetric changes followed the calculations of Warrick et al.,[49] for which the overall beach profile is assumed to remain constant over the decadal time scales provided by the satellite-derived shoreline data. Under these conditions, the volumetric change is equivalent to the product of the mean rate of shoreline change (m/yr), the beach segment length (m), and the total beach thickness (m), which is equivalent to the height over which sediment is spread between the depth of closure and the upper beach surface. We use a total beach thickness estimate of 12 m for southern California, which approximates the mean conditions of 8-to-10 m depth of closure and 2-to-4 m of beach elevation[49,79,80]. Uncertainty in these volumetric change calculations was assessed by considering a worst-case beach profile change, the largest sediment addition event from southern California's largest sediment source: the winter 2005 events at the Santa Clara River[53,54]. This event discharged approximately 5 million m³ of littoral grade sediment to the coast and dramatically altered the topography and bathymetry of beach profiles near the river mouth, including a massive alteration of the submarine longitudinal profile from the deposition of a 4–10 m thick submarine bar[53,54].

Under this condition with a highly unsteady beach profile, we find that our satellite-derived shorelines recorded an average of 66.3 m of shoreline widening from this event over the integrated 3.3 km beach segment length. Using the assumption of a 12 m thickness of the beach, we compute 2.6 million m³ of net volumetric gain in littoral sediment from the satellite-records, which is 52% of the actual amount (i.e., 48% error). This estimated volume is less than the actual measured volume owing to the abundance of sediment in the submarine portion of the profile, a bathymetric feature that the shoreline position measurements are unable to capture. Although our simple estimate is off by ~50%, we note that the submarine bar of sediment was eroded over the subsequent 5 yr, resulting in a beach profile that returned to its original shape, consistent with the assumptions of our calculations[53]. Regardless, even under the event-response conditions highlighted above, which capture a worst-case scenario during our shoreline position records, we find that our volume estimate is within 50% of the actual value. Using this example and the understanding that most of our shoreline data is derived from beaches without these types of morphodynamics, we have included ±50% total uncertainty for the shoreline-derived littoral sediment volumes.

## Data availability
Beach segment locations, source CoastSat transects, and mean annual shoreline positions and annual values of uncertainty for all 329 beach segments of California are tabulated and published in Warrick (2025)[74]. The raw shoreline position measurements that were used to derive the mean annual shoreline positions are published and available from the CoastSat data publication by Vos[42], version 1.5. Beach width measurements for each CoastSat transect are published in Warrick and Buscombe[76]. Hindcast wave data were obtained from the European Centre for Medium-Range Weather Forecasts (ECMWF) Reanalysis Version 5 (ERA5) hindcasts of as detailed by Soci et al.[61] and made available at https://www.ecmwf.int/en/forecasts/dataset/ecmwf-reanalysis-v5 (accessed May 14, 2025). Beach nourishment rates were obtained from the ASBPA National Beach Nourishment Database as detailed by Elko et al.[64] and made available at: https://asbpa.org/national-beach-nourishment-database/ (accessed May 14, 2025). Mean river discharge of littoral-grade sand for the Santa Barbara Littoral Cell was obtained from Barnard and Warrick[54].

## Code availability
Codes used to generate shoreline positions from satellite imagery are provided in the gitbub site: https://github.com/kvos/CoastSat. Codes used to compute deseasonalized annual shoreline positions from the raw CoastSat data are provided in the USGS Software Release by Warrick[44]: https://doi.org/10.5066/P1UYMOK7.

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

## Acknowledgements

This work was supported by the U.S. Geological Survey's Coastal and Marine Hazards and Resources Program through the Remote Sensing Coastal Change project. Additional funding provided by NASA grant NNH21ZDA001 (BS, TH). Any use of trade, firm, or product names is for descriptive purposes only and does not imply endorsement by the U.S. Government.

## Author contributions

J.A.W.—Primary author, data analyses, synthesis. K.V.—Data generation (shoreline change), data analyses. D.B.—Study design, data analysis, secondary author. A.R.—Study design, data analysis. S.V.—Data synthesis, secondary author. T.H.—Data generation and synthesis (longshore transport, sediment budgets). B.S.—Study design, primary author.

## Competing interests

The authors declare no competing interests.
