## [Transparent Peer Review file · Nature Communications]

Net Widening of Southern California Beaches

Corresponding Author: Dr Jonathan Warrick

Version 0:

Reviewer comments:

Reviewer #1

(Remarks to the Author)

Reviewer #2

(Remarks to the Author)

Net Widening of Southern California Beaches
Warrick et al

The authors describe a new method to analyze a 41yr satellite-derived shoreline time series of shoreline position estimates along the CA coast. The new method breaks the mobile sand-gravel portions of the the CA coastline into 330 segments of varying lengths delineated by natural and human coastal barriers that can block or significantly hinder longshore transport. Separating CA into 3 regions: south, central, and north CA using Pt Conception and SF Bay as regional boundaries. They find that the regional average shoreline change in southern CA (129 segments) is distinct from the other regions, with a statistically significant positive trend in beach width. Their segmented method reveals that roughly 2/3 of the southern CA regional average beach widening is owing to just 4 outlier (in a good way) segments, representing just 7% of the total length of southern CA's sandy coast. These segments have widened dramatically owing to a combination of beach nourishments and longshore drift retention by coastline barriers. More generally, they link the wide range of different segment changes with local deepwater and nearshore wave hindcast-driven cross-shore and longshore transport, historical nourishment and bypassing records, and sand budget estimates.

The paper is thorough, well written, and contains valuable new information for coastal scientists, engineers, managers and policymakers. I recommend publication but ask the authors to consider the following comments and possible revisions.

1. The title of the paper

The compact title leaves a first impression that southern CA beach widening, i.e. from an increase in overall beach volume (vs. volume-conserving beach rotations), has been widely distributed across the region, rather than primarily the result of a small number of rapidly widening segments. Given the more outlier-driven nature of the net volume increase-driven net widening, and the sand management importance of the longshore sand redistribution that has occurred throughout the region, perhaps a title like,

“Concentrated Beach Widening and Widespread Sand Redistribution in Southern California”

would speak more to the 2 important conclusions of the paper?

2. Beach areas

The paper favors a less common "beach area" metric over the more common use of mean beach width of mean shoreline position change, presumably because this is explicitly conserved when integrating over variable length new-method-segment results. Nevertheless, because area and width are simply related by $dArea = 100 * N_{transects} * dWidth$, as you state in the

paper, and I don't know what the heck a 2M m² change in social beach area really means for starters, it would be helpful if you gave both area and width quantities throughout in the text (eg. the beach width change in brackets next to the area or, to my taste, vice versa), and in the figures where possible (eg. as new right y-axes labels in Fig 3).

3. Total beach area

I may have missed it, but because you have back beach locations for all the CoastSat transects, and the latest version of CoastSat has better geospatial referencing, I presume you can guesstimate the total area and true mean beach width of social beaches? If so, you could also present that number and your area change numbers as a percent change of the total beach area, and actual mean beach width as well? Those would be interesting percentages to put the millions of square meters of change into context.

4. Segment math

The upside of segments is that they really get the reader to appreciate the length scales you have to pay attention to (short!) even when thinking about long-term beach behavior. A downside of using variable length shoreline segments as the basis of a regional analysis is the loss of explicit longshore shoreline change conservation with somewhat subjectively-driven alongshore averaging variability. Some of the segments are tiny, some are pretty long, so when you say things like 'X% of the segments widened' it has no real quantitative meaning as far as I can tell? I would avoid non-quantitative segment stats if you can, unless I am missing something. I worry about the little guy - the small segments losing their collective regional voice in the analysis. Do they add up to a significant part of the 320km? Are they preferentially dropped when you filter by segment trend significance in some parts of the analysis?

5. Time series start and end points

lines 119-124 state:

The net widening trend in southern California is much stronger than northern California, and it can be approximated using linear regression ($\pm 95\%$ c.i.) to be $+56,000 \pm 17,000$ m²/yr, which is equivalent to $+2.3 \pm 0.7$ million m² of beach area formation over the 41-yr measurement record. An endpoint approach provides similar results, for example, by comparing the first and last three years of the 41-year time series (i.e., 1984-1986 and 2022-2024), which results in an additional 2.25 million m² of new beach area across southern California.

--

I would avoid manipulating/multiyear averaging the ends of the time series if you can. It implies you think the annual values have too much uncertainty to be interpreted individually, like their uncertainty is worse than the regression thinks, but then treat regression p values < 0.05 as gospel.

Given that the southern CA-wide average annual beach area or beach width (ie. Figure 3i) can vary significantly on 4-5 year ENSO timescales and beyond, the "end point approach" result seems highly dependent on when your time series starts and ends? If you had written this paper in 2013-2015 (the max area year), then your end point beach area would have been fantastic. If your time series ended shortly after the 2016 El Nino, not so much. The trend vs end point result seems like a coincidence?

Being a devil's advocate with the end point game, imagine if the CoastSat time series had happened to started at the 1990 peak in Fig 3i (Landsat 1 launch delayed further!). We would be saying there has been no hint of a trend in the satellite record. I think that is why some smart people invented statistics - to keep us from playing end point games to feed our own natural biases. The real take home message is there has not been a statistically detectable LOSS of regionally averaged beach width since 1984, as many fools had assumed there would be in the PreVossian Era (my hand is raised...seems like that was the Stone Age now).

On the same uncertain end point topic, in Fig 3 you calculate the cumulative change relative to a 1984-1986 mean starting point. That will "lift up" the plots a little because CoastSat shows a regionally-averaged retreating shoreline in the first couple years, so you've suppressed the first dip a bit yes? Are any of your results truly sensitive to whether you do that or not? As you state clearly in the paper, the story has mostly been an amazing run of sand redistribution on O(few beach segment) space scales + 4 outlier positive segments thanks to us humans.

6. Trend averaging

lines 145-148 state:

Summing these beach segment trends using weighting by shoreline length produces a mean shoreline change rate of $+0.17$ m/yr, which is equivalent to $+6.9$ m of average beach widening over the integrated 320-km study area during the 41-yr records.

--

It is my understanding that you cannot legally add regression trends together, even with the segment length weighting.

$(\text{regress}(X)+\text{regress}(Y))/2$ is not generally equal to $\text{regress}((X+Y)/2)$? I'm unsure throughout the paper where and when you are integrating trends and when you are integrating shoreline segment time series before calculating a trend. The latter is certainly safer than the former even if they turn out to be very close in the end sometimes with correlated data?

7. Segment analysis

I did my best to understand how you derive a single annual shoreline position time series for each coastal segment. Probably no set of supplemental plots can easily convey what the methodology is up to all the time. So be it, if it makes things better. Are you able to show that it buys you something worthwhile for annual time series estimates over just data analysis 101 annual (Nov-Oct) time averaging of transects across a segment with cursory QC of the raw CoastSat time series (eg. matlab rmoutliers followed by linear interpolation gap filling)? At the larger littoral cell or southern CA averaging levels, if you calculate a regional mean time series from your new method segment single time series, and then the equivalent using all the original CoastSat transects reduced to the simple annual means, are they very different?

If answering this would require unreasonable additional work to answer, please just say so and disregard. I am not questioning the motivation for, or the veracity of, the new method. CoastSat errors on small time-space scales is a thing, but simpler annual averages start to quickly tame it when validated with surveys, so it would be useful to know in what situations and/or time-space scales the "loss of a simple sense of how annual-spatial means are being derived for a section of coastline" is worth the improved accuracy of a more bespoke data-taming methodology, but that can wait for another time.

8. Littoral Cell and All southern CA time series

Along with a righthand shoreline position change Y-axis, could you add a second color shading to Fig 3 that is just the 4 outlier segments in their appropriate littoral cells and then all in 3i? It would support your narrative that the 4 segments are a big part of the cumulative change.

9. Longshore Transport

Lines 564-579 state:

Potential sediment transport represents the volumetric flow rate of sand that is feasible given the available wave energy and is estimated using the CERC9 equation:

$$Q = K\gamma\sqrt{g}H^{5/2} \sin(2\theta b) \quad [3]$$

where Q is the transport potential, K is an empirical coefficient, g is the gravitational acceleration constant, H is the significant wave height, and θb is the approach angle of waves relative to the shore normal direction, ($\theta_{\text{shore normal}} - \theta_{\text{peak}}$).

--

I suggest that you scrap θ_{peak} (the mean wave direction at the peak period) and then scrape the CDIP-MOP "Bulk Sxy angle" (waveDm) from the netcdf files on the CDIP THREDDS server. θ_{peak} is biased towards swell directions in socal and has skewed your transport results. CERC longshore E_{flux} is closely related to $S_{xy_total} = E_{\text{total}} \sin(2 * (D_{\text{normal}} - \text{waveDm}))$. waveDm is much more consistent with what CERC Eq 3 θb wants to be. Given his publication history on longshore transport, I believe Dr. Vitousek will agree with this recommendation.

Reviewer #3

(Remarks to the Author)

The manuscript titled 'Net Widening of Southern California Beaches' is well written and presented and I found the science behind it rather solid. However, even though I have no major concerns about the quality of the research, I cannot recommend it for publication in NCOMMS. The reason is that I do not find the work presented to be relevant to the scope of a high IF journal like NCOMMS.

The manuscript discusses shoreline change dynamics in Southern California and tries to link the observations with possible drivers of change. That makes it a very interesting piece of work for a journal like Marine Geology, Geomorphology, Ocean and Coastal Management, JGR (among others), but not NCOMMS. The reason is that even though the work is solid, it lacks any feature that would make it stand out from the several other good papers out there. The spatial extent is quite limited, there are no new insights or knowledge presented that would be relevant to a broader audience (beyond SoCal stakeholders and a sub group of the coastal research community). The drivers of change are known, the methods used also rather standard and experienced colleagues could interpret the observations in similar fashion without most of the interesting and solid work done. Finally, the interpretation of the shoreline change is not so detailed and most of the important knowledge gaps remain open: what are the real drivers of the changes? Which temporal scales are more responsible for the overall trend; storms or long term effects? What is the relative importance of cross- vs long-shore transport, vs sediment sources and sinks?

Some more detailed comments on the methods:

- The authors imply that they estimate beach volumetric change by considering a constant beach profile in time and if I understood correctly also in space. But this assumption is rather weak in systems that undergo a constant change. There, mean profiles can change due to the overall trend, but also due the introduction of different sediments from beach nourishment. Finally, human interventions like spreading the sediment can make the beach wider without any volumetric

change involved.

- The authors seem to largely neglect cross-shore change which is important for the sediment budget.

Version 1:

Reviewer comments:

Reviewer #1

(Remarks to the Author)

The authors have carefully considered and addressed all comments and suggestions and have provided detailed responses to reviewers' comments. The authors have revised their assessment of uncertainty for their volumetric calculations and further emphasised the degree of uncertainty for these calculations in the revised manuscript providing a full justification for and explanation of its relevance. They have introduced substantial revisions, including extensively restructuring of the manuscript. The revised version strengthens the narrative and brings to light the broader implications and relevance to sediment management, adaptation strategies and coastal resilience. They have further provided additional detail in the methods section on their shoreline analysis approach which allows for replication of their method.

Overall, I find this is a manuscript of very high quality that represents a significant contribution in the field. In my view, it is ready for publication, and I am looking forward to seeing it published in due course.

Reviewer #2

(Remarks to the Author)

I thank the authors for considering my questions and suggestions thoughtfully. I look forward to the paper's publication.

A few final comments as the authors finalize their manuscript

1. The title of the paper

"Concentrated Widening of Southern California Beaches" seems like a better description of your revised manuscript and it's primary conclusion: that sand has been mostly migrating and widening/narrowing at sub-littoral cell scales owing to multiple factors. While I think this finding of the paper will "age well", I'm not so sure about the current title once the old assumption of net (as in almost-everywhere-net) beach loss over the last few decades is behind us. Net widening has a bait-and-switch feel to it even by the end of the abstract.

2. beach area vs. beach width, and beach expansion vs beach widening

It seems widening is used interchangeably with beach width and beach area, while expansion is used at times but only with beach area. $2D = \text{expansion} = \text{area}$ and $1D = \text{widening} = \text{width}$ would be the ideal, but technically area change in this case is really also =widening because the shoreline dimension is fixed. Perhaps it would be best to just stick with widening everywhere?

3. beach width plots

Thank you for adding in the supplementary section. I'll make one last lobbying effort to have you put the littoral cell beach width plots (Fig S2) in a 2nd column next to the beach area plots in Fig 3. I think they are far more interesting together because the cells have such different alongshore scales. eg. the Santa Barbara has a big net beach area signal, but a modest mean beach width signal, because it is a big cell.

minor Supp Fig S2 edit: The caption refers to r_{adj} correlation coefficients but there are none.

4. Plotting results relative to a three year 1984-86 mean starting point.

Thanks for explaining this did not influence the trend analysis. Might be good to say that when you describe what you are doing at lines 532-536? eg. "Lastly, for presentation and comparison purposes (but not in the trend analysis), the mean shoreline position...".

Introducing selective post-analysis interannual time averaging still comes across as a little under-motivated with the wording you have. And some of those early years have above average ERA5 wave power. If the early annual data is too spotty to make a stable starting year all by itself by your standards, why not start everything with the first year you think is stable? As you note, the overall trend analysis is not very sensitive to the starting details.

Reviewer #3

(Remarks to the Author)

I have carefully read the revised manuscript and the exchanges with the editor and the other reviewers. Unfortunately, my

overall opinion remains unchanged. While I consider this to be a solid piece of work that would have value if published as a technical report or in a specialized journal, I do not think it merits publication in Nature Communications. I recognize that the other reviewers hold different views, and I also asked the editor for guidance on how to proceed, but I did not receive a response. Ideally, I would have preferred to see the opinion of an additional colleague rather than revisiting the paper myself; nevertheless, I will provide my reply to the authors.

Novelty and significance

A main limitation of the work is that none of its central conclusions appear to be new to the community. The paper presents some marginal methodological refinements, but not genuine advances beyond what is already well established. The authors themselves acknowledge in their response that “the computation and use of SDS has become fairly mature [14,15],” but then argue that their use of signal processing tools yields “unique and unexpected findings.” I challenge the authors to clarify what is truly unique here, since the processes they describe are standard material in textbooks on coastal processes.

Scale and methodology

Regarding the small- vs. large-scale issue, I fully agree that local studies can be impactful. However, I do not believe this manuscript achieves that. In fact, it appears to combine the shortcomings of both scales:

Unlike strong local studies, the authors do not employ accurate site-specific data. Instead, they rely on stylized beach profiles and global ERA-5 reanalysis, rather than local measurements or regionally calibrated models. This is not far from the methodologies applied by some of the large scale studies the authors seem to be critical of.

Instead, I believe that the authors could have done much more. For a site as important as California—indeed recognized as foundational in littoral science—I would expect regionally tailored reanalyses and/or field data of waves ocean circulation and, ideally, repeated surveys (LiDAR, drones, or DGPS) that provide direct topographic information. These could demonstrate actual changes in beach morphology and volume and allow in depth interpretation. Without such data, reported increases in beach area could equally reflect shoreline accretion, bluff erosion, longshore transport, or beach nourishment—each of which carries very different implications for coastal processes.

In the absence of detailed local information, many of the apparent findings may simply result from the limited size of the study area and the relatively short observation record, which are not sufficient to suppress the influence of outliers. This is a well-known weakness of small-scale or short-term studies. For example, there is only one location, driving the overall accreting trend for the whole area. And (as discerned by the discussion with the other reviewers), adding one of more decades of observation could potentially result in a different, or even the opposite trend.

Review of the manuscript *Net Widening of Southern California Beaches*

I would like to comment the authors on their open approach to quantification of uncertainty in SDS and SDS-derived metrics. The authors apply a novel approach to SDS time series that minimizes the effects of sparse and irregular data, a common issue with satellite-based studies. They also perform a detailed and thorough estimation of possible sediment inputs/outputs to the Southern California littoral systems (within the limitation of available data) from different sources including beach nourishments, dredging, river inputs, and sediment transport and redistribution between cells.

Results highlight major gaps and challenges in the understanding and management of coastal systems and provide evidence of the opportunity to improve coastal management and develop more economically and environmentally effective coastal management solutions. Furthermore, although their approach relies on the availability of extensive and detailed datasets that may not be broadly available, the methodology followed and discussion provide the framework for assessing and optimizing coastal sediment management practices that can lead to more resilient coasts, more sustainable management interventions, and better use of public funds. This paper is well structured and well written, the methods are clear and replicable, and the figures produced are of very high quality and informative. This work is therefore highly relevant and suitable for publication with only minor editing.

Introduction

RV1. Referring to the statement: "... based on the decomposition analyses of Warrick et al.³¹ to reduce the effects of measurement errors, which are generally 10-15 m (1- σ) for synoptic observations³²" : Although the benchmarking paper reports accuracies in the order of 10-15m RMSE (except for one site), several studies have shown that these uncertainties are potentially higher, depending on coastal setting and latitude/geographical locations. Although this may not be the case for Southern California, this statement implies that SDS accuracy is 10-15 m in all environments around the globe. I therefore feel that a statement recognising this fact is warranted, but could be as simple as, for example: "... for synoptic observations in similar environments" or soothing to that effect.

Results

RV2. *Southern California Littoral cells*: The names of the different cells do appear in the Fig2 caption, but which cell is which is not evident to readers unfamiliar with the region. I was wondering if there is a way clarifying which section is which in the text without hindering the flow of this section?

RV3. “The combination of time and waves explains between 30 and 94% (mean = 65%) of the beach area variability at the littoral cell scale (Table 2).”. Earlier, you provide a value for the % variability wave power explains over the entire region (0 -51%; mean = 31%). I suggest also providing the equivalent for time alone here to enable the direct comparison between the three.

Methods

RV4. Referring to the statement: “segmentation of the scene into water, whitewater, and sand land cover”: I believe strictly speaking, segmentation is into water, whitewater, sand, and other land cover.

RV5. “...we find negligible differences in results from the different ERA5 hindcast locations.”: I appreciate you recognise that the ERA5 wave data used may not represent local wave conditions well. Do you have some understanding of where your sediment transport results based on ERA5 may be over(under) estimating transport rates? Where might any such differences be important if at all?

RV6. You make a commendable effort to account for uncertainty associated with the different elements of this work. However, the way uncertainty in volumetric change is calculated ($2\text{-}\sigma$ uncertainty in SDS trend*[beach thickness =3.0m], only accounts for uncertainty in shoreline change while introducing a new mostly arbitrary parameter. Changes in the beach profile whether interannual or consistent trends are not accounted for. Though I appreciate some level of subjective/informed decisions may be necessary in broadscale approaches and do not necessarily compromise the validity or value of results, some quantification of the variability and long-term trends in beach profiles and how these might influence results would be useful.

Discussion

RV7. “With the continued supply of sediment to southern California littoral cells from rivers and bluff erosion^{21,39,41,51}, continued exposure of the **coastal** to changing wave”. *Should it be coast* instead of coastal?

Figures

RV8. Fig4: I assume the dashed white line in 1-c represent the location for panels d-f? If yes, could you please include a statement in the caption? If not, please indicate location of d-f on a-c.

RV9. Fig3: please indicate location of d-f on a-c.

Supplementary material

RV10. Table S1: all values in the right hand column are NaNs...

RV11. Table S2: caption: "based on **the** ASBPA Beach Nourishment Database¹⁴ and additional insights from local records^{15–19}." ?

REVIEWER COMMENTS

Reviewer #1

I would like to comment the authors on their open approach to quantification of uncertainty in SDS and SDS-derived metrics. The authors apply a novel approach to SDS time series that minimizes the effects of sparse and irregular data, a common issue with satellite-based studies. They also perform a detailed and thorough estimation of possible sediment inputs/outputs to the Southern California littoral systems (within the limitation of available data) from different sources including beach nourishments, dredging, river inputs, and sediment transport and redistribution between cells.

Thank you for recognizing the unique approach to our data analyses. We are encouraged that one of the novel elements of this paper, and one of the reasons it may provide broad interest to the readers of Nature Communications, is this novel data analysis approach. As noted more fully below we have updated the Methods section to provide specific details of our analysis techniques, so they can be easily transferred to other sites throughout the world.

Results highlight major gaps and challenges in the understanding and management of coastal systems and provide evidence of the opportunity to improve coastal management and develop more economically and environmentally effective coastal management solutions. Furthermore, although their approach relies on the availability of extensive and detailed datasets that may not be broadly available, the methodology followed and discussion provide the framework for assessing and optimizing coastal sediment management practices that can lead to more resilient coasts, more sustainable management interventions, and better use of public funds. This paper is well structured and well written, the methods are clear and replicable, and the figures produced are of very high quality and informative. This work is therefore highly relevant and suitable for publication with only minor editing.

Thank you for these positive words.

Introduction

RV1. Referring to the statement: "... based on the decomposition analyses of Warrick et al.³¹ to reduce the effects of measurement errors, which are generally 10-15 m (1- σ) for synoptic observations³²" : Although the benchmarking paper reports accuracies in the order of 10-15m RMSE (except for one site), several studies have shown that these uncertainties are potentially higher, depending on coastal setting and latitude/geographical locations. Although this may not be the case for Southern California, this statement implies that SDS accuracy is 10-15 m in all environments around the globe. I therefore feel that a statement recognising this fact is warranted,

but could be as simple as, for example: “... for synoptic observations in similar environments” or soothing to that effect.

The reviewer is correct that there are sites for which SDS measurement uncertainties are greater than 10-15 m. For example, beaches with offshore bars that are exposed subaerially during lower tidal stages can have 10s of meters of shoreline position error. That noted, there are also locations, such as simple steep beaches, that have errors lower than the range we provided. Combined, this is one of the reasons that the original text used the phrase, “which are generally 10-15 m.” That noted, we have revised our statement to include a qualification for beach type as suggested. This should help with the understanding and application of these results.

Results

RV2. Southern California Littoral cells: The names of the different cells do appear in the Fig2 caption, but which cell is which is not evident to readers unfamiliar with the region. I was wondering if there is a way clarifying which section is which in the text without hindering the flow of this section?

Yes, great suggestion. We would like the littoral cells to be as clear as possible for the readers. We attempted to add the cell names to the Fig. 2, but the text would not fit in the space provided. Thus, we continue to number the cells in the figure and include the littoral cell names in the figure caption. Additionally, we have added littoral cell names and the cell numbers from Fig. 2 to Figure 3. We have also added the cell numbers to Figures 4 and 5. We hope that this will help ‘connect the dots’ between the cell numbers and names for readers.

RV3. “The combination of time and waves explains between 30 and 94% (mean = 65%) of the beach area variability at the littoral cell scale (Table 2).”. Earlier, you provide a value for the % variability wave power explains over the entire region (0 -51%; mean = 31%). I suggest also providing the equivalent for time alone here to enable the direct comparison between the three.

Yes, excellent suggestion. We have revised the manuscript to include a complete set of the correlation analysis results.

Methods

RV4. Referring to the statement: “segmentation of the scene into water, whitewater, and sand land cover”: I believe strictly speaking, segmentation is into water, whitewater, sand, and other land cover.

Yes, this is correct. We have corrected this error. Thank you for your careful reading of this section.

RV5. "...we find negligible differences in results from the different ERA5 hindcast locations.": I appreciate you recognise that the ERA5 wave data used may not represent local wave conditions well. Do you have some understanding of where your sediment transport results based on ERA5 may be over(under) estimating transport rates? Where might any such differences be important if at all?

This is an excellent question about the limitations of the ERA5 wave hindcast data. General understanding of the wave fields in the Southern California Bight is that they are influenced strongly by wave direction and topographic/bathymetric complexities throughout the region[6,7]. The ERA5 data do not incorporate these spatial complexities, but they were the only available dataset to compare with our 1984-2024 shoreline records.

The inadequacy of the ERA5 hindcast data to provide local-scale wave conditions was the primary reason that we did not use the beach segment data for our statistical comparisons. Rather, we used the integrated littoral cell and total integrated southern California region as comparative areas. Although we did communicate this adequately (given the reviewer's comment, see above), we did not effectively communicate the limitations of these comparisons. We have revised the presentation of this data source to include limitations, which include the effects of shoreline orientation and offshore topography and bathymetry on the relative levels and year-to-year variations of wave power delivered to the integrated shoreline of the littoral cells.

However, a complete quantitative analysis of these over(under) estimating patterns would be a unique research project that is far beyond the scope of this paper and would require investigations that include mode integration, calibration and validation, similar to the work that was required to develop the 2000-2025 CDIP wave hindcasts from other data sources by O'Reilly and others[7].

Thus, we have made the presentation clearer about the potential uncertainties in the available data, and we highlight our assumptions that the southern California littoral cells may include wave power variabilities that are caused by the region's geography.

RV6. You make a commendable effort to account for uncertainty associated with the different elements of this work. However, the way uncertainty in volumetric change is calculated (2- σ uncertainty in SDS trend[beach thickness =3.0m], only accounts for uncertainty in shoreline change while introducing a new mostly arbitrary parameter. Changes in the beach profile whether interannual or consistent trends are not*

accounted for. Though I appreciate some level of subjective/informed decisions may be necessary in broadscale approaches and do not necessarily compromise the validity or value of results, some quantification of the variability and long-term trends in beach profiles and how these might influence results would be useful.

Thank you for these suggestions and concerns. First, we acknowledge that the greatest changes to beach profiles will be over the seasonal scales and have characterized these patterns for California in recent papers[8,9]. Because of the high variability at seasonal scales, the focus for the manuscript in question was time scales over which these seasonal effects would be minimized.

The annual-scale data that were generated for our manuscript allow for a comparative perspective over annually integrated time scales. An annually integrated beach profile would eliminate seasonal variabilities, which are known to be statistically significant for at least 90% of California's beaches[8]. That is, we should expect that the year-to-year mean annual beach profiles (if they were consistently measured for a beach) would have more uniform shapes compared to the more variable seasonal profiles (such as bar evolution over the 'summer' vs. 'winter' profiles). This is the underlying basis for the first-order assumption that the beach profile shapes are approximately similar over the integrated annual increments of time used in the volume change estimates[10]. Clearly, this assumption has limitations because beach profile shapes are known to evolve over multi-year increments of time, especially in the evolution of submarine bars[11,12].

In light of this, we revised our assessment of uncertainty for the volumetric calculations by including a data analysis of a southern California location with the greatest known multi-year beach profile evolution. As detailed in the response to the Editor above, we show that this worst-case scenario from real beach profile data results in at most 50% error in our volume estimates. [The worst case is a massive sediment deposition event from the largest sediment supply in southern California, the Santa Clara River, during the largest recorded event that resulted in a massive alteration in the profile in the form of a submarine bar – these conditions are highly unusual and have not been met at any other California site during our 1984-2024 records[4,5]]. Thus, we report these calculations in the revised manuscript and revise the total uncertainty of our estimates to this 50% level, even though the actual uncertainty is likely much lower for most beach segments (i.e., annual-scale profile shape changes are likely considerably lower than the massive changes to the river mouth sediment bar highlighted in our worst-case scenario).

Lastly, we include further characterization and justification of the parameters used for the volume estimates in the revised manuscript, and we provide new descriptions if not simple warnings in the manuscript (as noted in the response to the Editor) about the weakness of these estimates. For example, we provide

information and citations about the use of 12 m for the typical range of beach thicknesses (i.e., subaerial beach heights of 2-4 m plus depth of closure estimates of 8-10 m, results in total vertical thickness of the active beach of ~12 m). That noted, the resulting littoral sediment volume estimates from the shoreline positions are useful for order-of-magnitude comparisons with other sediment budget calculations, and as such they serve as only one (albeit a weak one) of several lines of evidence for our conclusions.

Discussion

RV7. “With the continued supply of sediment to southern California littoral cells from rivers and bluff erosion^{21,39,41,51}, continued exposure of the coastal to changing wave”. Should it be coast instead of coastal?

Yes, thank you. Changes made.

Figures

RV8. Fig4: I assume the dashed white line in 1-c represent the location for panels d-f? If yes, could you please include a statement in the caption? If not, please indicate location of d-f on a-c.

Thank you for catching this. We described the white dashed lines in the text of the manuscript, but not the figure caption. This is fixed. The lines show the geographic extent of the sediment budget calculations.

RV9. Fig3: please indicate location of d-f on a-c.

We believe the reviewer is referring to Fig. 4, not Fig. 3. If so, these locations are added with letters on the seaward sides of the beaches.

Supplementary material

RV10. Table S1: all values in the right hand column are NaNs...

Thank you for identifying this potential problem. However, all the northern and central California values were NaNs, the southern California values were 1-8 (denoting the littoral cells; bottom of the table). That said, these data have been moved into the published USGS Data Product, which can be found here: <https://doi.org/10.5066/P139YRVP>. Because this new data publication captures all of the data in Table S1, the table was removed from the Supplemental Materials.

RV11. Table S2: caption: “based on the ASBPA Beach Nourishment Database¹⁴ and additional insights from local records^{15–19}.”?

Yes, thank you. Corrections made.

Reviewer #2

The authors describe a new method to analyze a 41yr satellite-derived shoreline time series of shoreline position estimates along the CA coast. The new method breaks the mobile sand-gravel portions of the the CA coastline into 330 segments of varying lengths delineated by natural and human coastal barriers that can block or significantly hinder longshore transport. Separating CA into 3 regions: south, central, and north CA using Pt Conception and SF Bay as regional boundaries. They find that the regional average shoreline change in southern CA (129 segments) is distinct from the other regions, with a statistically significant positive trend in beach width. Their segmented method reveals that roughly 2/3 of the southern CA regional average beach widening is owing to just 4 outlier (in a good way) segments, representing just 7% of the total length of southern CA's sandy coast. These segments have widened dramatically owing to a combination of beach nourishments and longshore drift retention by coastline barriers. More generally, they link the wide range of different segment changes with local deepwater and nearshore wave hindcast-driven cross-shore and longshore transport, historical nourishment and bypassing records, and sand budget estimates.

The paper is thorough, well written, and contains valuable new information for coastal scientists, engineers, managers and policymakers. I recommend publication but ask the authors to consider the following comments and possible revisions.

Thank you.

1. The title of the paper. The compact title leaves a first impression that southern CA beach widening, i.e. from an increase in overall beach volume (vs. volume-conserving beach rotations), has been widely distributed across the region, rather than primarily the result of a small number of rapidly widening segments. Given the more outlier-driven nature of the net volume increase-driven net widening, and the sand management importance of the longshore sand redistribution that has occurred throughout the region, perhaps a title like, "Concentrated Beach Widening and Widespread Sand Redistribution in Southern California" would speak more to the 2 important conclusions of the paper?

Thank you for this suggestion. We appreciate the reviewer's concern about being as clear as possible in the title. Although we tried to develop a better title, co-author consensus was that the existing title was both compact and thoroughly complete. The key to our decision was the use of "net" in the existing title:

"Net Beach Widening of Southern California Beaches"

Because it is both the first word of the title (giving it prominence) and the defining concept of the work (an integration across the beaches of the study area).

*2. Beach areas. The paper favors a less common "beach area" metric over the more common use of mean beach width of mean shoreline position change, presumably because this is explicitly conserved when integrating over variable length new-method-segment results. Nevertheless, because area and width are simply related by $dArea = 100 * N_{transects} * dWidth$, as you state in the paper, and I don't know what the heck a $2M m^2$ change in social beach area really means for starters, it would be helpful if you gave both area and width quantities throughout in the text (eg. the beach width change in brackets next to the area or, to my taste, vice versa), and in the figures where possible (eg. as new right y-axes labels in Fig 3).*

Yes, we agree that the use of area (sq. meters) is somewhat unconventional in coastal studies. One way to calibrate your eye to these numbers is by using a beach towel conversion factor (~2 sq. meters), which would suggest that the 2 million sq. meters of new beach area has increased total potential beach usage by ~ 1 million sunbathers. Similar conversion factors could be used for plover nesting sites or other ecological factors using their areal footprints.

The strength of using beach area is the ability to combine results across beach segments and littoral cells with simple addition. That is, if Littoral Cell A expanded by 500,000 sq.m. and Cell B expanded by 200,000 sq.m., then there is a total of 700,000 sq.m. of new beach area in the combined cells. Units of beach width expansion across these two cells cannot be combined by simple addition, because they need to be weighted by the beach length of each cell. Thus, there is a simplicity in the use of beach area, if you can transition toward this type of unit.

To help readers with the area units, however, we have revised the results to include both area and average beach width (i.e., length) metrics as suggested by the reviewer. Figures 1 and 3 have been replicated in the Supplemental Materials but converted into units of average length. These added figures are noted in the figure captions and the main text. Additionally, we have revised the text by including several more conversions of the areal change results into average beach width changes. These are found through the Results section. We hope this makes the results easier to understand and compare with other studies.

3. Total beach area. I may have missed it, but because you have back beach locations for all the CoastSat transects, and the latest version of CoastSat has better geospatial referencing, I presume you can guesstimate the total area and true mean beach width of social beaches? If so, you could also present that number and your area change numbers as a percent change of the total beach area, and actual mean beach width as well? Those would be interesting percentages to put the millions of square meters of change into context.

Excellent suggestion. Yes, we do have these number and have added the calculations and results to the text as suggested. In southern California there was ~23 million sq. m of beach in 1984, and our measurements suggest that this area expanded by ~10% (or 2.3 million sq.m) over the pending 41 years. This addition to the results should help readers understand the magnitude of the coastal responses.

4. Segment math. The upside of segments is that they really get the reader to appreciate the length scales you have to pay attention to (short!) even when thinking about long-term beach behavior. A downside of using variable length shoreline segments as the basis of a regional analysis is the loss of explicit longshore shoreline change conservation with somewhat subjectively-driven alongshore averaging variability. Some of the segments are tiny, some are pretty long, so when you say things like 'X% of the segments widened' it has no real quantitative meaning as far as I can tell? I would avoid non-quantitative segment stats if you can, unless I am missing something. I worry about the little guy - the small segments losing their collective regional voice in the analysis. Do they add up to a significant part of the 320km? Are they preferentially dropped when you filter by segment trend significance in some parts of the analysis?

Another excellent suggestion. We agree that any presentation of the number or percentage of beach segments may provide misleading information. We have revised the manuscript and removed all discussion of the number or percentages of the beach segments. All results are presented in units of the total beach length (km). For example, we now state that, “beach segments with significant widening trends ($p < 0.05$) represent 155 km of the 320-km total shoreline length (49%) assessed in our analyses of southern California.” Thank you for helping us improve the data presentation.

5. Time series start and end points

lines 119-124 state: “The net widening trend in southern California is much stronger than northern California, and it can be approximated using linear regression ($\pm 95\%$ c.i.) to be $+56,000 \pm 17,000$ m²/yr, which is equivalent to $+2.3 \pm 0.7$ million m² of beach area formation over the 41-yr measurement record. An endpoint approach provides similar results, for example, by comparing the first and last three years of the 41-year time series (i.e., 1984-1986 and 2022-2024), which results in an additional 2.25 million m² of new beach area across southern California.”

I would avoid manipulating/multiyear averaging the ends of the time series if you can. It implies you think the annual values have too much uncertainty to be interpreted individually, like their uncertainty is worse than the regression thinks, but then treat regression p values < 0.05 as gospel.

Yet another excellent suggestion. We agree wholly, and we revised the manuscript and removed all endpoint approaches to calculation of the trends. All trends are based on linear regressions and the related statistics.

Given that the southern CA-wide average annual beach area or beach width (ie. Figure 3i) can vary significantly on 4-5 year ENSO timescales and beyond, the “end point approach” result seems highly dependent on when your time series starts and ends? If you had written this paper in 2013-2015 (the max area year), then your end point beach area would have been fantastic. If your time series ended shortly after the 2016 El Nino, not so much. The trend vs end point result seems like a coincidence?

Yes, good point. This was another important reason that we removed all of the endpoint trend analyses and results from the revised paper. Thank you for helping us see that these endpoint calculations were a weakness of the data analyses.

Being a devil’s advocate with the end point game, imagine if the CoastSat time series had happened to started at the 1990 peak in Fig 3i (Landsat 1 launch delayed further!). We would be saying there has been no hint of a trend in the satellite record. I think that is why some smart people invented statistics - to keep us from playing end point games to feed our own natural biases. The real take home message is there has not been a statistically detectable LOSS of regionally averaged beach width since 1984, as many fools had assumed there would be in the PreVossian Era (my hand is raised...seems like that was the Stone Age now).

As the reviewer properly notes, trend results will be highly dependent on the time intervals evaluated. In fact, one can find statistically significant intervals of widening and narrowing in the integrated southern California data, if short increments of time are considered. For example, the 2005-2010 interval is one with statistically significant beach narrowing for southern California, whereas 2010-2015 is significantly widening (see Fig. 3i). This is one of the reasons that we used linear regression on the entire data set for trend computations. We note that these time series likely did not start on a low point, because some of the storms of record were during the 1982-'83 winter, whereas our data start in mid-1984, an interval of time for which we see significant recovery of beach width (e.g., compare with other storm-recovery intervals, such as 1998-1999, 2010-2011 and 2016-2017; Fig. 3i).

We wanted to explore the worst-case scenarios that the reviewer provides (e.g., a start date during a peak beach wide point like 1990). Southern California beach area trends continue to be positive (i.e., widening) and statistically significant in this case. For the 1990 start date, the trend ($57,100 \text{ m}^2/\text{yr}$) is slightly greater than the trend for the entire data ($56,300 \text{ m}^2/\text{yr}$), although the difference is not

significant at the 95% c.i. (btw... more on start date effects in the next response note). Thus, we agree that the statistical techniques are the most valuable in evaluating changes in these shoreline data, largely owing to the metrics of goodness of fit and uncertainty that can help readers understand how to interpret the underlying data. And these results reinforce the removal of the endpoint trend calculations from the manuscript.

On the same uncertain end point topic, in Fig 3 you calculate the cumulative change relative to a 1984-1986 mean starting point. That will “lift up” the plots a little because CoastSat shows a regionally-averaged retreating shoreline in the first couple years, so you’ve suppressed the first dip a bit yes? Are any of your results truly sensitive to whether you do that or not? As you state clearly in the paper, the story has mostly been an amazing run of sand redistribution on O(few beach segment) space scales + 4 outlier positive segments thanks to us humans.

This is an important point. Thank you for raising it. Please note that the data are plotted with respect to the initial condition (1984-1986) set to zero, which was done to provide a standard and comparative initial point. However, the trends are computed with linear regression, for which the y-axis offset does not influence the resulting slope. That is, the initial condition of the shoreline positions could be set to -1000 meters (or +1000 meters), and the linear slope of the time series would be the same. Our calculation of the total change in beach area is based wholly on the linear regression slope multiplied by the interval of time (41 years). Thus, initial conditions will not influence that result.

The reviewer brings up an important point about the sensitivity to dips (and rises) in the data, especially during the initial conditions (late 1980s). We would note that if Landsat 5 had been launched 1 or 2 years earlier, we would have captured the high erosion winter of 1982-83[13], which would have included a deeper dip in the earliest portion of the record. Thus, we can assume that the trends would have been even higher if our records started in 1982 or 1983 owing to a larger potential ‘dip’ effect.

To evaluate the potential scale of this effect, we tested the sensitivity of the initial year used for the record and computing linear regression slopes for the southern California data. These results are tabulated here:

Initial Year	Linear Trend (m ² /yr)	Percent of Original Trend
1984	56,392	100%
1985	54,978	97.5%
1986	56,465	100.1%
1987	55,831	99.0%
1988	56,157	99.6%
1989	54,974	97.5%

1990	57,123	101.3%
1991	62,589	111.0%
1992	59,143	104.9%
1993	55,310	98.1%

Thus, the range of computed trends varies marginally (-2.5% to 11%) with different initial years. This provides increased confidence that any perceived dips in the initial southern California records have minor (at best) effects on the trend results.

6. *Trend averaging. lines 145-148 state: “Summing these beach segment trends using weighting by shoreline length produces a mean shoreline change rate of +0.17 m/yr, which is equivalent to +6.9m of average beach widening over the integrated 320-km study area during the 41-yr records.”*

It is my understanding that you cannot legally add regression trends together, even with the segment length weighting. $(\text{regress}(X)+\text{regress}(Y))/2$ is not generally equal to $\text{regress}((X+Y)/2)$? I’m unsure throughout the paper where and when you are integrating trends and when you are integrating shoreline segment time series before calculating a trend. The latter is certainly safer than the former even if they turn out to be very close in the end sometimes with correlated data?

Thank you for bringing this up. I hindsight we realize that we did not communicate our methods adequately, and in response, we have revised the Methods section and descriptions in the Results.

In the end, we are only providing trends calculated from single time series records. For littoral cells, we have combined all the representative beach segments into a single time series record for the integrated cell. This time series record is then analyzed for trends using linear regression. Similarly, an evaluation of the combination of multiple littoral cells is provided by first developing a single integrated time series of beach area for these cells and then analyzing by linear regression. Thus, the combination of regression trends was not conducted by summation of regression results.

We apologize that this was unclear in the previous version of the manuscript. It should be much clearer in the revised manuscript.

7. *Segment analysis. I did my best to understand how you derive a single annual shoreline position time series for each coastal segment. Probably no set of supplemental plots can easily convey what the methodology is up to all the time. So be it, if it makes things better. Are you able to show that it buys you something worthwhile for annual time series estimates over just data analysis 101 annual (Nov-*

Oct) time averaging of transects across a segment with cursory QC of the raw CoastSat time series (eg. matlab rmoutliers followed by linear interpolation gap filling)? At the larger littoral cell or southern CA averaging levels, if you calculate a regional mean time series from your new method segment single time series, and then the equivalent using all the original CoastSat transects reduced to the simple annual means, are they very different?

If answering this would require unreasonable additional work to answer, please just say so and disregard. I am not questioning the motivation for, or the veracity of, the new method. CoastSat errors on small time-space scales is a thing, but simpler annual averages start to quickly tame it when validated with surveys, so it would be useful to know in what situations and/or time-space scales the "loss of a simple sense of how annual-spatial means are being derived for a section of coastline" is worth the improved accuracy of a more bespoke data-taming methodology, but that can wait for another time.

This is a very important point. Given the reviewer's comment about an inability to replicate the methods, we have completed updated the Methods section to include a step-by-step process for how the mean annual shoreline position values were calculated. We believe that this technique should be relatively easy to follow now, but if it is not, please point out how it is unclear. We would like our analysis technique to be something others can mimic, test, and modify as necessary, and a clear description of it is essential.

Regarding the comparison with other techniques, we do see this as somewhat of a big ask, however, we have provided new information about why we developed our 'deseasonalize' technique. Key elements of the techniques include: (i) the reduction in scatter that may be introduced from the seasonality of the shoreline position (highlighted in Supp. Fig. S9) and (ii) the removal of seasonal bias from undersampling in the original data (highlighted in Supp. Fig. S10). The reason that this latter effect is important can be seen in the large amount of undersampling in the pre-2000 data when only one satellite (Landsat 5) was collecting data. During this era, there was commonly whole seasons without data collection (e.g., the Eureka cell had no winter season data for most years, as did many of the rest of California during years; Supp. Fig. S10). If there was no data collected during the 4-6 months with the narrowest beach conditions, then mean annual values would be biased high (wide). Removing the regular seasonal component of the time series is a standard method – and the best one we can find – for removing this kind of potential bias.

8. Littoral Cell and All southern CA time series. Along with a righthand shoreline position change Y-axis, could you add a second color shading to Fig 3 that is just the 4 outlier segments in their appropriate littoral cells and then all in 3i? It would support your narrative that the 4 segments are a big part of the cumulative change.

Thank you for this suggestion. We tried modifying Fig. 3, but adding lines made for a relatively busy plot. To help with the communication of the important of the rapidly widening beach segments, we have revised the text to include new descriptions and comparisons of the data. We hope that these help the reader understand the magnitude of these effects.

9. Longshore Transport. Lines 564-579 state:

“Potential sediment transport represents the volumetric flow rate of sand that is feasible given the available wave energy and is estimated using the CERC9 equation:

$$Q = K\gamma\sqrt{gH^5/2} \sin(2\theta b) [3]$$

where Q is the transport potential, K is an empirical coefficient, g is the gravitational acceleration constant, H is the significant wave height, and θb is the approach angle of waves relative to the shore normal direction, ($\theta_{shore\ normal} - \theta_{peak}$).”

*I suggest that you scrap θ_{peak} (the mean wave direction at the peak period) and then scrape the CDIP-MOP “Bulk Sxy angle” (waveDm) from the netcdf files on the CDIP THREDDS server. θ_{peak} is biased towards swell directions in social and has skewed your transport results. CERC longshore E_{flux} is closely related to $Sxy_{total} = E_{total} * \sin(2 * (D_{normal} - waveDm))$. waveDm is much more consistent with what CERC Eq 3 θb wants to be. Given his publication history on longshore transport, I believe Dr. Vitousek will agree with this recommendation.*

Absolutely. Thank you! Corrections made and Methods, Figures and Supplemental Materials have all been updated with the revised calculations. You will note subtle differences in the littoral transport directions.

Reviewer #3:

The manuscript titled 'Net Widening of Southern California Beaches' is well written and presented and I found the science behind it rather solid. However, even though I have no major concerns about the quality of the research, I cannot recommend it for publication in NCOMMS. The reason is that I do not find the work presented to be relevant to the scope of a high IF journal like NCOMMS.

The manuscript discusses shoreline change dynamics in Southern California and tries to link the observations with possible drivers of change. That makes it a very interesting piece of work for a journal like Marine Geology, Geomorphology, Ocean and Coastal Management, JGR (among others), but not NCOMMS. The reason is that even though the work is solid, it lacks any feature that would make it stand out from the several other good papers out there. The spatial extent is quite limited, there are no new insights or knowledge presented that would be relevant to a broader audience (beyond SoCal stakeholders and a sub group of the coastal research community). The drivers of change are known, the methods used also rather standard and experienced colleagues could interpret the observations in similar fashion without most of the interesting and solid work done. Finally, the interpretation of the shoreline change is not so detailed and most of the important knowledge gaps remain open: what are the real drivers of the changes? Which temporal scales are more responsible for the overall trend; storms or long term effects? What is the relative importance of cross- vs long-shore transport, vs sediment sources and sinks?

Thank you for your assessments of how this research fits into the broader context of coastal science. Although these conclusions differ markedly with Reviewers #1 and #2, we respect the struggle this reviewer had with fitting a regionally based study into the subset of science that is globally important. Obviously, there are many ways that this importance could be examined and evaluated.

We would suggest that several elements help make our work relevant to a broader audience. First, we provide novel analysis tools for evaluating a new a growing data set (satellite-derived shorelines, or SDS), and these tools can be applied worldwide. Although the computation and use of SDS has become fairly mature[14,15], there have not been significant advances in the analysis tools to better interpret these SDS data. In fact, several top coastal colleagues continue to complain openly that SDS data are too noisy and therefore unusable. Our techniques cut through that noise of SDS data using novel signal processing tools, resulting in findings that are unique and unexpected.

Second, although California is a small region of the world's coast, it has served as an important site in the development of foundational littoral science[16–19]. Thus, it is important to highlight when there is change to this foundational understanding, which we argue has occurred in our work.

Third, we would caution readers that studies must be ‘global in scale’ to be relevant to the broader scientific community. We will highlight that a series of ‘global scale’ studies of shoreline change that have been published in the top tier of scientific journals[20–22], because all of these articles have been called into serious question by the research community[23–25]. This suggests that although the SDS data have become global in scale, coastal researchers have not developed the appropriate tools to evaluate these data, and the ‘global’ results that have been published in a series of high impact papers may have either insignificant or erroneous conclusions. The point being that the geographic extent of a study should play a secondary control on its relevance. Rigor and novelty should be the ultimate metric of relevance. For example, some of the most influential and recent geoscience research in high-impact journals has come from small-scale-to-regional studies[26–29].

Some more detailed comments on the methods:

- The authors imply that they estimate beach volumetric change by considering a constant beach profile in time and if I understood correctly also in space. But this assumption is rather weak in systems that undergo a constant change. There, mean profiles can change due to the overall trend, but also due the introduction of different sediments from beach nourishment. Finally, human interventions like spreading the sediment can make the beach wider without any volumetric change involved.

Thank you for this comment. We have completely revised the presentation, uncertainty analysis and use of these computations as noted in the responses to the Editor and Reviewers. Your comments were consistent with the others, so we will not repeat the details of these changes here.

- The authors seem to largely neglect cross-shore change which is important for the sediment budget.

We have included a detailed analysis of cross-shore exchange at the annual scale across the littoral cells. We do not need to address seasonal-scale cross-shore exchanges – which are both large in the study area and something we have published on recently[8,9] – owing to the annualization of the shoreline position data. That is, our technique for computing mean annual shoreline positions removes these seasonal effects. In the end, we find that waves can explain a significant amount of the cross-shore exchange of sediment (as represented in the shoreline position[18]), but that these cross-shore exchanges do not explain the underlying time-dependent trends in the data. These results are summarized in the statistical results shown in Table 2, which have not been changed in any manner in the revisions here.

REFERENCES CITED:

1. Warrick, J.A., and Milliman, J.D. (2003) Hyperpycnal sediment discharge from semiarid southern California rivers: Implications for coastal sediment budgets. *Geology*, **31** (9), 781.
2. Warrick, J.A., and Farnsworth, K.L. (2009) Sources of sediment to the coastal waters of the Southern California Bight, in *Earth Science in the Urban Ocean: The Southern California Continental Borderland*, Geological Society of America.
3. Inman, D.L., and Jenkins, S.A. (1999) Climate Change and the Episodicity of Sediment Flux of Small California Rivers. *The Journal of Geology*, **107** (3), 251–270.
4. Barnard, P.L., and Warrick, J.A. (2010) Dramatic beach and nearshore morphological changes due to extreme flooding at a wave-dominated river mouth. *Marine Geology*, **271** (1–2), 131–148.
5. Warrick, J.A. (2020) Littoral Sediment From Rivers: Patterns, Rates and Processes of River Mouth Morphodynamics. *Front. Earth Sci.*, **8**, 355.
6. Adams, P.N., Inman, D.L., and Graham, N.E. (2008) Southern California Deep-Water Wave Climate: Characterization and Application to Coastal Processes. *Journal of Coastal Research*, **244**, 1022–1035.
7. O'Reilly, W.C., Olfe, C.B., Thomas, J., Seymour, R.J., and Guza, R.T. (2016) The California coastal wave monitoring and prediction system. *Coastal Engineering*, **116**, 118–132.
8. Warrick, J.A., Buscombe, D., Vos, K., Kenyon, H., Ritchie, A.C., Harley, M.D., Janda, C., L'Heureux, J., and Vitousek, S. (2025) Shoreline Seasonality of California's Beaches. *JGR Earth Surface*, **130** (2), e2024JF007836.
9. Warrick, J.A., Buscombe, D., Vos, K., Ritchie, A.C., and Battalio, B. (2025) Seasonal rotation of California pocket beaches. *Earth Surf Processes Landf*, **50** (8), e70115.
10. Warrick, J.A., Vos, K., Buscombe, D., Ritchie, A.C., and Curtis, J.A. (2023) A Large Sediment Accretion Wave Along a Northern California Littoral Cell. *JGR Earth Surface*, **128** (7), e2023JF007135.
11. Ruggiero, P., Kaminsky, G.M., Gelfenbaum, G., and Cohn, N. (2016) Morphodynamics of prograding beaches: A synthesis of seasonal- to century-scale observations of the Columbia River littoral cell. *Marine Geology*, **376**, 51–68.
12. Stevens, A.W., Ruggiero, P., Parker, K.A., Vitousek, S., Gelfenbaum, G., and Kaminsky, G.M. (2024) Climate controls on longshore sediment transport and coastal morphology adjacent to engineered inlets. *Coastal Engineering*, **194**, 104617.

13. Dingler, J.R., and Reiss, T.E. (2002) Changes to Monterey Bay beaches from the end of the 1982–83 El Niño through the 1997–98 El Niño. *Marine Geology*, **181** (1–3), 249–263.
14. Vos, K., Harley, M.D., Splinter, K.D., Simmons, J.A., and Turner, I.L. (2019) Sub-annual to multi-decadal shoreline variability from publicly available satellite imagery. *Coastal Engineering*, **150**, 160–174.
15. Vitousek, S., Buscombe, D., Vos, K., Barnard, P.L., Ritchie, A.C., and Warrick, J.A. (2023) The future of coastal monitoring through satellite remote sensing. *Camb. prisms Coast. futures*, **1**, e10.
16. Inman, D.L., and Chamberlain, T.K. (1960) Littoral sand budget along the southern California coast. 245–246.
17. Inman, D.L., and Brush, B.M. (1973) The Coastal Challenge. *Science, New Series*, **181** (4094), 20–32.
18. Yates, M.L., Guza, R.T., and O’Reilly, W.C. (2009) Equilibrium shoreline response: Observations and modeling. *Journal of Geophysical Research*, **114** (C9).
19. Komar, P.D., and Inman, D.L. (1970) Longshore sand transport on beaches. *J. Geophys. Res.*, **75** (30), 5914–5927.
20. Voudoukas, M.I., Ranasinghe, R., Mentaschi, L., Plomaritis, T.A., Athanasiou, P., Luijendijk, A., and Feyen, L. (2020) Sandy coastlines under threat of erosion. *Nat. Clim. Chang.*, **10** (3), 260–263.
21. Almar, R., Boucharel, J., Graffin, M., Abessolo, G.O., Thoumyre, G., Papa, F., Ranasinghe, R., Montano, J., Bergsma, E.W.J., Baba, M.W., and Jin, F.-F. (2023) Influence of El Niño on the variability of global shoreline position. *Nat Commun*, **14** (1), 3133.
22. Nienhuis, J.H., Ashton, A.D., Edmonds, D.A., Hoitink, A.J.F., Kettner, A.J., Rowland, J.C., and Törnqvist, T.E. (2020) Global-scale human impact on delta morphology has led to net land area gain. *Nature*, **577** (7791), 514–518.
23. Cooper, J.A.G., Masselink, G., Coco, G., Short, A.D., Castelle, B., Rogers, K., Anthony, E., Green, A.N., Kelley, J.T., Pilkey, O.H., and Jackson, D.W.T. (2020) Sandy beaches can survive sea-level rise. *Nat. Clim. Chang.*, **10** (11), 993–995.
24. Warrick, J.A., Buscombe, D., Vos, K., Bryan, K.R., Castelle, B., Cooper, J.A.G., Harley, M.D., Jackson, D.W.T., Ludka, B.C., Masselink, G., Palmsten, M.L., Ruiz De Alegria-Arzaburu, A., Sénéchal, N., Sherwood, C.R., Short, A.D., Sogut, E., Splinter, K.D., Stephenson, W.J., Syvitski, J., and Young, A.P. (2024) Coastal shoreline change assessments at global scales. *Nat Commun*, **15** (1), 2316.

25. Zăinescu, F., Anthony, E., Vespremeanu-Stroe, A., Besset, M., and Tătui, F. (2023) Concerns about data linking delta land gain to human action. *Nature*, **614** (7947), E20–E25.
26. Trimble, S.W. (1997) Contribution of stream channel erosion to sediment yield from an urbanizing watershed. *Science*, **278**, 1442–1444.
27. Barksdale, M.B., Hein, C.J., and Kirwan, M.L. (2023) Shoreface erosion counters blue carbon accumulation in transgressive barrier-island systems. *Nat Commun*, **14** (1), 8425.
28. Ohenhen, L.O., Shirzaei, M., Ojha, C., Sherpa, S.F., and Nicholls, R.J. (2024) Disappearing cities on US coasts. *Nature*, **627** (8002), 108–115.
29. Vacchi, M., Shaw, T.A., Anthony, E.J., Spada, G., Melini, D., Li, T., Cahill, N., and Horton, B.P. (2025) Sea level since the Last Glacial Maximum from the Atlantic coast of Africa. *Nat Commun*, **16** (1), 1486.

Editors

RE: Nature Communications manuscript NCOMMS-25-54103A

28 Nov 2025

Dear Editors,

Thank you for providing detailed peer reviews and your own assessments of our Nature Communications manuscript, "**Net Widening of Southern California Beaches.**" Your decision on the manuscript was to invite **additional revisions**. We are pleased to provide our revised manuscript and this point-by-point response to the review comments.

In the materials below, you will find the original review comments in *italics font* and our responses and manuscript changes in regular font. We look forward to our continued collaboration with you on this review process.

Sincerely,

Jonathan A. Warrick, PhD

U.S. Geological Survey

EDITOR:

Thank you again for submitting your revised manuscript "Net Widening of Southern California Beaches" to Nature Communications. We have now received reports from the reviewers who evaluated the original version. On the basis of their comments (copied below), we have decided to invite an additional revision of your work.

You will see that, while the reviewers find that your revisions improved the manuscript, some important points remain to be addressed. We would like the authors to emphasise the novelty and contribution of their work, not only in the Abstract, but particularly in the Discussion. With regards to field data, would they have changed the results have the authors considered field data in their study? This is a relevant point that should be addressed in the response letter and within the main text. Please revise your manuscript, addressing all the remaining issues raised by the reviewers.

Thank you for this synthesis and these suggestions. As recommended, we have included revised Abstract, Introduction and Discussion sections to include more emphasis on the novelty of this work. The focus on these revisions is that the most heavily urbanized and dammed section of study area (southern California) has seen the largest gains in beach area. This novelty stems from the fact that these kinds of land use changes generally result in reductions of sediment supply and increases in coastal erosion on a global basis, therefore, the southern California region is truly unique. Although these findings were already part of the manuscript, we give them more emphasis in the revised manuscript, which should assist with relating the overall novelty of the work.

The request for further consideration of the available field data is also relevant to a study such as ours, but unfortunately there is a lack of regular (i.e., seasonal) coastal topography of the beaches in the study area. Exceptions include a few beaches in the San Diego region (SIO studies) and Ocean Beach near San Francisco (USGS study). These beaches represent ~2.5% of the California beach shoreline length, and the available time series data are only 30-50% as long as the satellite-based records. Additionally, our recent work (Warrick et al., 2025, JGR-ES) suggests that these beaches have unique shoreline change patterns compared to the diversity of coastal settings in the study area, so they cannot be used for 'representative' study sites for California or even their littoral cells. Thus, we use these field data for uncertainty analyses (for which our analysis techniques do extremely well – see Methods section and Supplemental Materials Figures). We have made these details clearer in the opening paragraphs of the revised manuscript, where we directly address the availability of field data for assessments.

Lastly, as noted below, we have addressed the complete set of reviewer comments. Thank you for this opportunity to resubmit this study to Nature Communications.

REVIEWER COMMENTS

Reviewer #1 (Remarks to the Author):

The authors have carefully considered and addressed all comments and suggestions and have provided detailed responses to reviewers' comments. The authors have revised their assessment of uncertainty for their volumetric calculations and further emphasised the degree of uncertainty for these calculations in the revised manuscript providing a full justification for and explanation of its relevance. They have introduced substantial revisions, including extensively restructuring of the manuscript. The revised version strengthens the narrative and brings to light the broader implications and relevance to sediment management, adaptation strategies and coastal resilience. They have further provided additional detail in the methods section on their shoreline analysis approach which allows for replication of their method.

Overall, I find this is a manuscript of very high quality that represents a significant contribution in the field. In my view, it is ready for publication, and I am looking forward to seeing it published in due course.

Thank you. We look forward to seeing this work published too.

Reviewer #2 (Remarks to the Author):

I thank the authors for considering my questions and suggestions thoughtfully. I look forward to the paper's publication.

A few final comments as the authors finalize their manuscript

1. The title of the paper

“Concentrated Widening of Southern California Beaches” seems like a better description of your revised manuscript and it's primary conclusion: that sand has been mostly migrating and widening/narrowing at sub-littoral cell scales owing to multiple factors. While I think this finding of the paper will “age well”, I'm not so sure about the current title once the old assumption of net (as in almost-everywhere-net) beach loss over the last few decades is behind us. Net widening has a bait-and-switch feel to it even by the end of the abstract.

The suggestion is to change the term “net” to “concentrated” in the title. While both terms provide accurate descriptions of our findings, their meanings are quite different.

The adjective ‘net’ may be defined as: “remaining after the deduction of all charges, outlay, or loss” or “excluding all nonessential considerations: basic, final.” [Merriam-Webster dictionary, 2025]. Please note that this definition is not consistent with the suggested definition of this term, which necessitates all case-

by-case (or beach-by-beach) results to be consistent for the use of the term 'net.' For this definition of 'net' to be true, every beach would have to show net widening independently before the term could be applied. This situation is not only highly improbable, especially for a large study area such as ours, but it is also inconsistent with the most common usage and definitions of the word 'net.' For example, if a company shows a "net profit", this does not mean that all units/projects of the company were profitable. Rather, the term is understood to mean that overall the sum of the finances was net positive. A consistent application is used here, the net beach area has increased, even though there is a diversity of beach trends within the study area.

The adjective 'concentrated' may be defined as: "contained or existing or happening together in a small or narrow space or area." The challenge with the adjective 'concentrated' is that it provides no understanding/information about the magnitude of the noun in question (i.e., beach change). Rather, it simply denotes that the change has a limited spatial extent. In contrast, the word 'net' provides information about the overall summation of change.

We contend that the total (or sum of the) change is a more important result than the spatial extent/variability of the changes. To limit reader confusion on our use of 'net' (which we portend is completely consistent with its definition), please note that the Abstract clearly states that "...several beaches experienced severe erosion..." and that "...sediment is not effectively distributed to vulnerable beaches." These statements provide clear information that our results are in no way "almost-everywhere-net" as the reviewer contends. As such, we disagree that there is any hint of "bait-and-switch" in our presentation.

In the end, the title will remain because it is accurate and captures the most important results of our work. We have added a date range to the title, however, for clarity.

2. beach area vs. beach width , and beach expansion vs beach widening

It seems widening is used interchangeably with beach width and beach area, while expansion is used at times but only with beach area. 2D=expansion=area and 1D=widening=width would be the ideal, but technically area change in this case is really also =widening because the shoreline dimension is fixed. Perhaps it would be best to just stick with widening everywhere?

Thank you for this suggestion. We have given a detailed review and assessment of word usage and agree with the reviewer that consistent terms should be used. Because we will cite both beach area and beach width results in the paper, we have revised the paper to use 'widening' to represent changes to beach width and 'growth' to represent changes to beach area.

3. beach width plots

Thank you for adding in the supplementary section. I'll make one last lobbying effort to have you put the littoral cell beach width plots (Fig S2) in a 2nd column next to the beach area plots in Fig 3. I think they are far more interesting together because the cells have such different alongshore scales. eg. the Santa Barbara has a big net beach area signal, but a modest mean beach width signal, because it is a big cell.

Thank you for this suggestion. We provided the average beach width data and plots in the Supplemental as suggested during the previous round of reviews, however, there seems to be adequate interest in having these data side by side. Although these plots are somewhat redundant (the beach area data are divided by a constant, the total beach length, to result in average beach width), we have created a double panel figure that has both data sets. This revised figure is provided in the revised manuscript as Fig. 3 and the Supplemental Fig S2 has been removed.

minor Supp Fig S2 edit: The caption refers to r_{adj} correlation coefficients but there are none.

Thank you for catching this. This figure was integrated into Fig. 3, so the review comment is no longer applicable to the manuscript. R-adj values are provided in Fig. 3.

4. Plotting results relative to a three year 1984-86 mean starting point.

Thanks for explaining this did not influence the trend analysis. Might be good to say that when you describe what you are doing at lines 532-536? eg. "Lastly, for presentation and comparison purposes (but not in the trend analysis), the mean shoreline position..."

Introducing selective post-analysis interannual time averaging still comes across as a little under-motivated with the wording you have. And some of those early years have above average ERA5 wave power. If the early annual data is too spotty to make a stable starting year all by itself by your standards, why not start everything with the first year you think is stable? As you note, the overall trend analysis is not very sensitive to the starting details.

We have revised the introduction to these data to include a statement of the effect of start date on trend analyses. This includes a summary of the data analyses presented in the previous round of reviews, which shows that start date has a negligible impact on the results and is now included as Supp. Table S1. This should provide the necessary clarification about the sensitivity to start date.

Unfortunately, the remaining comments are somewhat conflicting. On one hand, it is concluded that, “the overall trend analysis is not very sensitive to the starting details” as we detailed in the previous review. But then, it is suggested that we should “start everything with the first year you think is stable.” In response to this, it could be asked: If it does not matter which year is the first, why should we revise the data presentation and change the first year?

This type of revision would require an adequate definition of a “stable year” for the California study area, but the relative stability of the shorelines varies geographically. For example, our findings suggest that there are multiple causes of shoreline change in the data (erosion during large storms, multi-year recovery between storms, large differences in storm exposure across the study area, widening following river sediment input and beach nourishment, longshore convergence of sediment, sediment accretion waves and erosional waves moving along the coast, etc.). These factors result in a wide diversity of time-dependent patterns in shoreline change (e.g., refer to the beach segment time series data plotted in the Supplemental Materials Fig. S2, S3, S4, S6 and S7).

Thus, it seems justified to use the most complete data available (1984-2024) for overall completeness and a revised (and better) description of the implications of start dates in trend results, which is now included in the text and a table in the Supplemental Materials.

Reviewer #3 (Remarks to the Author):

I have carefully read the revised manuscript and the exchanges with the editor and the other reviewers. Unfortunately, my overall opinion remains unchanged. While I consider this to be a solid piece of work that would have value if published as a technical report or in a specialized journal, I do not think it merits publication in Nature Communications. I recognize that the other reviewers hold different views, and I also asked the editor for guidance on how to proceed, but I did not receive a response. Ideally, I would have preferred to see the opinion of an additional colleague rather than revisiting the paper myself; nevertheless, I will provide my reply to the authors.

Thank you for providing a summary of this opinion, which as noted differs from the other two reviewers and the editor. A key element of this comment is the finding that the paper does not “merit publication in Nature Communications” even though the paper is described to be a “solid piece of work.”

These comments are difficult to respond to, largely because there are no specific details about what would and would not merit publication in Nature Communications. The Nature portfolio does provide some guidance in its Aims and Scope statements for each journal, but these details were not addressed.

Thus, we have not been provided with the grounds for which merit was not achieved.

As detailed above in the response to the Editor, we will note that we modified the Abstract, Introduction and Discussion to provide a better description of the novelty and utility of the study. These new materials and emphases should provide a stronger case for the merit of this work.

Novelty and significance

A main limitation of the work is that none of its central conclusions appear to be new to the community. The paper presents some marginal methodological refinements, but not genuine advances beyond what is already well established. The authors themselves acknowledge in their response that “the computation and use of SDS has become fairly mature [14,15],” but then argue that their use of signal processing tools yields “unique and unexpected findings.” I challenge the authors to clarify what is truly unique here, since the processes they describe are standard material in textbooks on coastal processes.

A central comment is that “none of (the) central conclusion appear to be new to the community,” but this statement does not provide specific details about which conclusions are being referred to. There are several conclusions of the paper, ranging from site and regionally specific findings to overall processes that dictate coastal change patterns. It would have been helpful to know which of these conclusions were being referred to. There is also no evidence for making this statement about the redundancy of the conclusions. For example, it would have been helpful to have citations of peer-reviewed papers that have the same/similar findings or redundant results from the study area. With more specific information, we would have been able to address the comments that we have not provided “genuine advances beyond what is already well established.” That is, without further information we cannot guess what is meant by “what is already established.” Perhaps, it is longshore transport? Or, the effects of large wave events? Or, the accumulation of remote sensing measurements to produce net beach area change results in the study area? Or, cumulative coastal change patterns and effects in the California study area?

In contrast, we contend that we provide new and novel findings. For example, our paper provides references from several studies that have concluded the opposite findings of ours (i.e., that southern CA beaches are experiencing “chronic erosion”). Our paper shows that there has been net beach widening. We also show how the southern California region contrasts with global-scale understanding about the impacts of dams on rivers. This unique finding occurs for southern California because of the sediment management, coastal development, natural sediment supply, and natural longshore transport in this system. Additionally, we

provide a new data analysis approach (deseasonalization), which to our understanding is unique and useful to the coastal sciences. Thus, our conclusion is that our paper is substantially different from others and therefore new to the existing literature.

To help readers further understand the newness and uniqueness of the work we have included revisions of the Abstract, Introduction, and Discussion to highlight the impacts of dams and urbanization generally across coastal watersheds throughout the world and how the results of this study are contrary to this collective understanding. This is not a new element of the manuscript, because it was highlighted in the Discussion previously, but it is emphasized more thoroughly in the revised manuscript to help readers understand additional elements of uniqueness of the study.

It is also noted that we state that “the computation and use of SDS has become fairly mature,” which is used to suggest that the results that we develop could not be new or unique. Two different things are being conflated. The first is “computation and use of SDS” and the latter is “new understanding of coastal processes and patterns.” The maturity of the development and use of SDS data (a measurement tool) does not preclude insights from these data. On the contrary, new insights are being developed regularly from the fairly mature SDS tools. For example, over the past couple of years and with the use of SDS data, our research group has discovered:

- Massive sediment accretion waves related upland watershed land use impacts (Warrick et al., 2023, JGR-ES).

- Spatially variable seasonal shoreline patterns, including great diversity in the timing of seasonal beach narrowing, which is dependent on coastal geography (Warrick et al., 2025, JGR-ES)

- Two dominant and spatially independent patterns in seasonal beach rotation along California (Warrick et al., 2025, ESPL).

- Net beach widening along the most heavily dammed and urbanized section of the California coast (this paper).

Many other groups are discovering similar unique coastal patterns and processes through the use of the recently developed SDS data (e.g., Bishop-Taylor et al., 2023; Vitousek et al, 2023; Vos et al., 2023; O’Reilly et al., 2025)

Thus, we would suggest that maturity of the SDS techniques is a prerequisite for new and novel observations of coastal morphodynamics. That is, the measurement tools need to be mature before new and novel understanding can be built with these tools. This is consistent with a broad set of sciences (medicine, climate, genetics, physics, etc.); for which measurement techniques must be mature before novel findings/understanding can be made. There are many

sources of information about the relationships between measurement tools and scientific advancements, perhaps the most approachable would be the 10-episode BBC series, “The Day the Universe Changed” or the materials of the Harvard University’s Collection of Historical Scientific Instruments (<https://chsi.harvard.edu/>).

Thus, we disagree with the contention that our findings cannot be unique because our measurement methods (i.e., SDS) are relatively mature, and we provide details of how our observations are unique.

Scale and methodology

Regarding the small- vs. large-scale issue, I fully agree that local studies can be impactful. However, I do not believe this manuscript achieves that. In fact, it appears to combine the shortcomings of both scales:

Unlike strong local studies, the authors do not employ accurate site-specific data. Instead, they rely on stylized beach profiles and global ERA-5 reanalysis, rather than local measurements or regionally calibrated models. This is not far from the methodologies applied by some of the large scale studies the authors seem to be critical of.

First, we would like to be clear about large-scale studies: We cite many global scale studies in our work (such as Vos et al. 2023, Luijendijk et al., 2018, and Syvitski et al. 2007) that have included rigorous analysis and detailed treatment of large data sets. These studies are essential to understanding physical processes, morphodynamics, and the diversity of patterns around the world. However, we, like many of our coastal colleagues, are highly critical of other global-scale studies that make conclusions from data poor statistical basis, have inadequate sampling resolution, or include incorrect baseline assumptions (Cooper et al., 2020, Nature Climate Change; Warrick et al., 2024, Nature Communications). Thus, we appreciate global-scale studies, especially those that provide scientific rigor, uncertainty assessments, and novel synthesis.

Second, there is the suggestion that our paper only relied on “stylized beach profiles and global ERA-5 reanalysis” and that it did not include “local measurements or regionally calibrated models”. This is incorrect. Specifically, we used local measurements from regionally calibrated models (the CDIP wave reanalysis of waves along the inner continental shelf of California), shoreline positions and orientations at 100-m shoreline spacing, a sediment transport model, and a novel scaling attribute, and the Peclet Number to evaluate the implications of longshore sediment transport patterns. These analyses were conducted at the local scale (100-m spacing along the entire southern California study area). These accurate and site-specific analyses are characterized in the Methods, Results, Discussion sections, detailed in the Supplemental Material

section, and referred to in the Abstract. These small-scale longshore transport calculations provide critical findings, which is one of the reasons that they are highlighted in the Abstract.

In addition to these site-specific measurements, we also provided several other local to regional analyses, including historical littoral sediment budgets, sediment volumetric calculations, and storm-related change assessments using ERA-5 data (the latter two of which are referred to in the review comment, but the techniques include many other input datasets than beach profiles and ERA5 wave assessments). Thus, we would suggest that in combination, we have done much more work and present much more data than “the large-scale studies the authors seem to be critical of.”

Instead, I believe that the authors could have done much more. For a site as important as California—indeed recognized as foundational in littoral science—I would expect regionally tailored reanalyses and/or field data of waves ocean circulation and, ideally, repeated surveys (LiDAR, drones, or DGPS) that provide direct topographic information. These could demonstrate actual changes in beach morphology and volume and allow in depth interpretation. Without such data, reported increases in beach area could equally reflect shoreline accretion, bluff erosion, longshore transport, or beach nourishment—each of which carries very different implications for coastal processes.

As noted above, we did much more than characterized in these comments, including the specific reanalysis and wave analyses asked for. These detailed analyses are important elements of the findings of our work and were (and still are) highlighted in the Abstract.

Regarding repeated field surveys, we utilized the complete set of data available for the study area, which represents only ~2.5% of the total shoreline length. As noted by Warrick et al. (2025) JGR-ES these field study sites are generally not representative of the diversity of shoreline settings and time-dependent change dynamics of the California coast. Furthermore, these field data represent only 30-50% of the time duration available from the satellite records. For these reasons, the field data were used only to verify the satellite data; they could not be relied upon for littoral-scale analyses of increases/decreases in beach area.

Similarly, there are limited lidar surveys of the complete study area (1998, 2009-2011, and 2016). Although these data are useful in computing long-term cliff erosion (Swirad and Young, 2022, Geomorphology), they have marked differences in the season of collection, which results in an inability to apply the lidar data to morphodynamic patterns or trends of the beaches. That is, seasonal and event-based patterns in the lidar data dominate the signals (for example, the 2016 lidar survey was designed to capture the extent of large-scale erosion

immediately following a year with big storm impacts), resulting in linear trends inconsistent with more detailed field or satellite-based observations. Thus, we unfortunately do not have an adequate lidar data set for beach morphodynamics across the study area.

Also, we provided analyses of the major causes of beach width changes including accretion, longshore transport and beach nourishment with independent assessments and calculations. This contrasts with the assessment that we were unable to identify the causes of beach widening. The paper provided tables, figures and supplemental information that summarized these causes of coastal change. Cliff erosion is not a factor for the widening beaches as these sites are largely along urbanized coastal sections.

Thus, we included (and continue to include) a wide range of data and analyses that were asked for in these comments.

In the absence of detailed local information, many of the apparent findings may simply result from the limited size of the study area and the relatively short observation record, which are not sufficient to suppress the influence of outliers. This is a well-known weakness of small-scale or short-term studies. For example, there is only one location, driving the overall accreting trend for the whole area. And (as discerned by the discussion with the other reviewers), adding one of more decades of observation could potentially result in a different, or even the opposite trend.

We will direct readers to our detailed response to Reviewer No. 2 comments in the previous round of reviews (and the brief reference to this work in the comments to Reviewer #2 above). We analyzed these data, and we found that measurement intervals and other factors were not important in the data and the computed trends. The spatial concern about “only one location driving the overall accreting trend” is inconsistent with the mathematics of summation and measurements of areas. Each beach segment is a summation of the 100-m spaced transects with the segment, and each littoral cell is the summation of the segments. The southern CA results are a summation of the region’s littoral cells. Uncertainty of each of these elements and summations are provided at the 2-sigma level (data published and available from the USGS as noted in the manuscript), so the influence of a single outlier location would be flagged by high levels of uncertainty, resulting in a summation that was not significantly different than zero. We note that these high levels of uncertainty do not occur in the results (e.g., Fig. 1 and 2).

In the end, we provide the most comprehensive (spatially and temporary) study of the broader California coast and its southern California region to date. Previous studies (e.g., Hapke et al., Amrouni et al., Yates et al., Kahl et al., Barnard et al. (2x), and Ludka et al.,) analyze much shorter/less complete time

series over much shorter lengths of coast. Thus, we can conclude that our study was certainly not “small-scale or short-term” when compared to existing studies.